# Physics-Informed Deep B-Spline Networks

**Zhuoyuan Wang**                                                                 *zhuoyuaw@andrew.cmu.edu*
*Department of Electrical and Computer Engineering*
*Carnegie Mellon University*

**Raffaele Romagnoli**                                                                 *romagnolir@duq.edu*
*Department of Mathematics and Computer Science*
*Duquesne University*

**Saviz Mowlavi**                                                                 *mowlavi@merl.com*
*Mitsubishi Electric Research Laboratories*

**Yorie Nakahira**                                                                 *yorie@cmu.edu*
*Department of Electrical and Computer Engineering*
*Carnegie Mellon University*

**Reviewed on OpenReview:** *https://openreview.net/forum?id=tHO2zEqmzm*

## Abstract

Physics-informed machine learning offers a promising framework for solving complex partial differential equations (PDEs) by integrating observational data with governing physical laws. However, learning PDEs with varying parameters and changing initial conditions and boundary conditions (ICBCs) with theoretical guarantees remains an open challenge. In this paper, we propose physics-informed deep B-spline networks, a novel technique that approximates a family of PDEs with different parameters and ICBCs by learning B-spline control points through neural networks. The proposed B-spline representation reduces the learning task from predicting solution values over the entire domain to learning a compact set of control points, enforces strict compliance to initial and Dirichlet boundary conditions by construction, and enables analytical computation of derivatives for incorporating PDE residual losses. While existing approximation and generalization theories are not applicable in this setting—where solutions of parametrized PDE families are represented via B-spline bases—we fill this gap by showing that B-spline networks are universal approximators for such families under mild conditions. We also derive generalization error bounds for physics-informed learning in both elliptic and parabolic PDE settings, establishing new theoretical guarantees. Finally, we demonstrate in experiments that the proposed technique has improved efficiency-accuracy tradeoffs compared to existing techniques in a dynamical system problem with discontinuous ICBCs and can handle nonhomogeneous ICBCs and non-rectangular domains. Code is available at https://github.com/jacobwang925/PI-BSNet.

## 1 Introduction

Recent advances in scientific machine learning have significantly accelerated progress in solving complex partial differential equations (PDEs). Physics-informed neural networks (PINNs) are proposed to combine information of available data and the governing physics model to learn the solutions of PDEs (Raissi et al., 2019; Han et al., 2018). However, in dynamics scenarios, both the parameters of the PDE and those defining initial and boundary conditions (ICBCs) can vary over time. Consequently, efficiently solving PDEs across a broad range of parameter values becomes essential. Learning solutions for families of parametric PDEs—especially when ICBCs are highly variable or discontinuous—remains a challenging task, requiring methods that are accurate, efficient, and theoretically sound. For example, in safety-critical control applications, time-varying

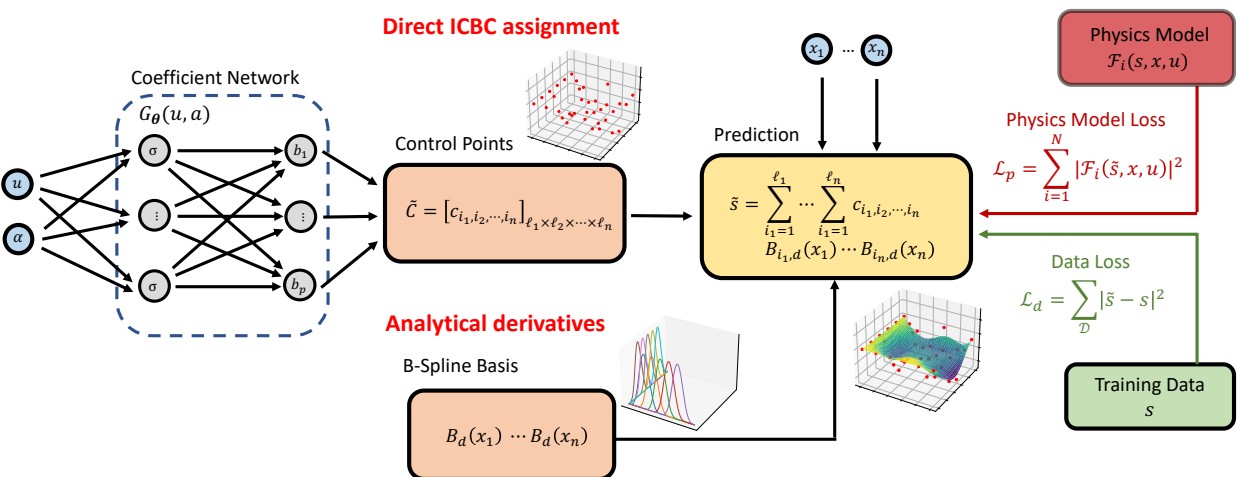

Figure 1: Diagram of PI-BSNet. The coefficient network takes system and ICBC parameters as input and outputs the control points tensor, which is then multiplied by the B-spline basis to produce the final output. Physics and data losses are used to train the coefficient network using closed-form derivative formulas, while the compliance of ICBC conditions is strictly enforced through the B-spline basis.

system dynamics and shifting safe regions result in continuously changing PDEs that characterize safety probabilities (Wang et al., 2025). However, solving these PDEs in real time with limited onboard resources is often prohibitively costly. This motivates the need for a neural network trained efficiently to represent a family of PDEs that enables rapid online inference of solutions under varying parameters and ICBCs.

Previous literature has generalized PINNs for parametric PDEs (Cho et al., 2024; Boudec et al., 2024; Huang et al., 2022), and operator learning methods have been developed to map input functions to PDE solutions (Kovachki et al., 2023; Li et al., 2020; Lu et al., 2019). These approaches typically impose initial and boundary conditions (ICBCs) as soft constraints through penalty terms in the loss function, which often require extensive tuning and do not guarantee strict compliance (Son et al., 2023; Brecht et al., 2023). Recent works have introduced PINNs with hard ICBC constraints by explicitly constructing solution ansatzes that satisfy the constraints by design (Wang et al., 2023; Li et al., 2024a; Chen et al., 2023; Liu et al., 2022; Sun et al., 2024). Such ansatzes are typically tailored to specific ICBCs and may lead to reduced accuracy in the interior (e.g., Liu et al. (2022); Sun et al. (2024), EPINN in Fig. 2). Moreover, most existing methods aim to directly learn PDE solutions on the entire domain (Lu et al., 2019; Cuomo et al., 2022), without exploiting more compact or structured representations. In such approaches, the PDE residuals in the training loss are computed by applying automatic differentiation over the entire space–time domain, which can be computationally-heavy. In addition, while many approximation and generalization bounds are derived for PINNs and neural operators (Kovachki et al., 2023; Lu et al., 2019; De Ryck & Mishra, 2022a;b; Mishra & Molinaro, 2023; 2022), theoretical results remain limited for families of PDEs represented using neural networks with structured bases.

**Contribution.** In this work, we propose a novel framework, termed physics-informed deep B-spline networks (PI-BSNet), that integrates neural B-splines and physics-informed learning to jointly solve families of parametric PDEs with different ICBCs (Fig. 1). Specifically, PI-BSNet is composed of B-spline basis functions and a parameterized coefficient network that learns the weights for the B-spline basis. The coefficient network takes inputs of the PDE and ICBC parameters, and outputs a tensor of control points (i.e., weights for the B-spline basis), which are then multiplied by the B-spline basis to generate the PDE values. This structured representation **enforces compliance of ICBCs by construction.** Training is performed using both physics-based and data-based loss functions to ensure the network accurately approximates PDE solutions across the domain for any PDEs in the parametric family. Furthermore, we show that the deep B-spline network serves as a **universal approximator for families of parametric PDEs** under mild assumptions, and we establish **generalization error bounds** for PI-BSNet. Finally, we present experimental results

showing that PI-BSNet has **improved computation vs. accuracy trade-offs** and is capable of handling nonhomogeneous ICBCs and non-rectangular domains. To the best of our knowledge, this is the first work to integrate B-spline basis representations with physics-informed learning for solving families of parametric PDEs while also providing theoretical guarantees on approximation accuracy and generalization error bounds.

## 2  Related Work

**Physics-Informed Learning.** Physics-informed neural networks (PINNs) are neural networks that are trained to solve supervised learning tasks while respecting physics laws described by partial differential equations (Raissi et al., 2019; Han et al., 2018; Cuomo et al., 2022). These approaches have been widely applied across domains, such as power systems (Misyris et al., 2020), fluid mechanics (Cai et al., 2022), and medical applications (Sahli Costabal et al., 2020), among others. Many variants of PINN have been developed to meet diverse learning requirements. For example, parametrized PINNs are proposed to solve parametric PDEs (Cho et al., 2024; Huang et al., 2022; de Avila Belbute-Peres et al., 2021; Qin et al., 2022; Lei et al., 2024). PINNs with hard constraints are proposed to meet specific ICBC requirements (Wang et al., 2023; Li et al., 2024a; Chen et al., 2023). PINN with adaptive rescaling or coupled differentiation schemes are proposed to reduce the computation burden in training (Ko & Park, 2025; Chiu et al., 2022). Under certain assumptions, PINNs are shown to exhibit bounded generalization error (De Ryck & Mishra, 2022a;b; Mishra & Molinaro, 2023; 2022) and convergence to ground truth solutions (Fang, 2021; Pang et al., 2019; Jiao et al., 2021).

Physics-informed neural operators (PINOs) are neural operators (Kovachki et al., 2023; Li et al., 2020; Lu et al., 2019; 2021; 2022) that learn mappings between functional spaces while respecting underlying physics laws. PINOs have been applied to both fixed and parametric PDEs (Wang et al., 2021b; Gao et al., 2021; Li et al., 2024b; Goswami et al., 2023), and extended to handle specialized problems such as pattern formation (Li et al., 2023) and varying ICBCs (Kumar et al., 2024). These methods typically impose ICBCs through loss terms, which may not guarantee strict compliance (Brecht et al., 2023).

In comparison, our method learns B-spline control point representations rather than solution values across the entire domain as in PINN/PINO approaches (e.g., Raissi et al. (2019); Han et al. (2018); Kovachki et al. (2023)). Moreover, our method directly enforces ICBCs through the B-spline structure itself, rather than through loss functions (e.g., Cuomo et al. (2022); Li et al. (2024b); Goswami et al. (2023)) or problem-specific filters (e.g., Wang et al. (2023); Li et al. (2024a); Chen et al. (2023)). These architectural differences yield better compliance with boundary conditions (Fig. 2), stable and fast training (Fig. 3), and improved tradeoff between computational efficiency and prediction accuracy (Fig. 4).

**B-splines and neural networks.** B-splines are piece-wise polynomial functions derived from slight adjustments of Bezier curves, aimed at obtaining polynomial curves that tie together smoothly (Ahlberg et al., 2016). B-splines have been integrated with finite element methods (Jia et al., 2013; Shen et al., 2023), employed in variational dual formulations for PDEs (Sukumar & Acharya, 2024), used to parameterize PDE domains (Falini et al., 2023), and adapted into spline-inspired mesh movement networks for PDEs (Song et al., 2022).

The combination of B-splines with neural networks has produced diverse applications, including surface reconstruction (Iglesias et al., 2004), nonlinear system modeling (Yiu et al., 2001; Wang et al., 2022b), image segmentation (Cho et al., 2021), and control system design (Chen et al., 2004; Deng et al., 2008). Kolmogorov–Arnold Networks (KANs) (Liu et al., 2024) employs spline functions to generate learnable weights as an alternative to traditional multilayer perceptrons. The neural network in our proposed PI-BSNet can take arbitrary MLP/non-MLP-based architectures, including KANs. In the regime of physics-informed learning, convolutional neural networks (CNNs) with Hermite spline kernels are trained with PDE and boundary condition loss functions to provide forward-time prediction in fixed-parameter PDEs (Wandel et al., 2022). Such method requires retraining when PDE parameters or boundary conditions change. In comparison, our work learns solutions on the entire state-time domain for parametric PDE families, and imposes hard compliance for varying ICBCs.

Closest to our work are methods that learn B-spline weights for fixed-parameter ODEs (Fakhoury et al., 2022; Romagnoli et al., 2024) and PDEs (Doległo et al., 2022; Zhu et al., 2024). Our work extends these approaches in several key directions. First, while these works focus on learning a single PDE with fixed ICBCs, we focus on joint learning of a family of parametric PDEs with different ICBCs with new theoretical guarantees (Section 4). Second, we leverage analytical derivatives of B-splines to enable efficient physics-informed learning. The use of physics-informed learning with hard ICBCs compliance enables accurate generalization beyond regions with available training data (Theorem 4.6 and prediction on unseen parameters in Section 5).

## 3 Proposed Method

### 3.1 Problem Formulation

The goal of this paper is to efficiently estimate high-dimensional surfaces governed by physics laws of a wide range variety of parameters (e.g., the solution of a family of PDEs). We denote $s : \mathbb{R}^n \to \mathbb{R}$ as the ground truth, i.e., $s(x)$ is the value of the surface at point $x$, where $x \in \mathbb{R}^n$. We assume the physics laws can be written as

$$\begin{aligned}
\mathcal{F}(s, x, u) = 0, & \ x \in \Omega(\alpha), \\
\mathcal{B}(s, x, u) = 0, & \ x \in \partial\Omega(\alpha),
\end{aligned} \tag{1}$$

where $\mathcal{F}$ is the physics law and $\mathcal{B}$ is the initial and boundary conditions (ICBCs), $u \in \mathcal{U} \subset \mathbb{R}^m$ is the parameters of the systems, $\Omega(\alpha) \in \mathbb{R}^n$ parameterized by $\alpha \in \mathcal{A} \subset \mathbb{R}^k$ is the domain of interest. Here, $\mathcal{U}$ and $\mathcal{A}$ ranges for the system and domain parameters and are bounded. In this paper, we consider $n$-dimensional bounded domain $\Omega = [a_1, b_1] \times [a_2, b_2] \times \cdots \times [a_n, b_n]$.[1] Our goal is to generate $\tilde{s}$ with neural networks to estimate $s$ on the entire domain of $\Omega$, with all possible parameters $u$ and $\alpha$. For example, in the case of solving 2D heat equations on $(x_1, x_2) \in [0, \alpha]^2$ at time $t \in [0, 10]$ with varying coefficient $u \in [0, 2]$ and $\alpha \in [3, 4]$, we have

$$\begin{aligned}
\mathcal{F}(s, x, u) = \partial s / \partial t - u \left( \partial^2 s / \partial x_1^2 + \partial^2 s / \partial x_2^2 \right) = 0, & \qquad x = (x_1, x_2, t) \in \Omega_x \times \Omega_t, & (2) \\
\mathcal{B}(s, x, u) = s - 1 = 0, & \qquad x = (x_1, x_2, t) \in \partial\Omega_x \times \Omega_t, & (3)
\end{aligned}$$

where $\Omega_x = [0, \alpha]^2$ and $\Omega_t = [0, 10]$, and $\partial\Omega_x$ is the boundary of $\Omega_x$. Here, equation 2 is the heat equation and equation 3 is the boundary condition. In this case, we want to solve for $s$ on $\Omega = \Omega_x \times \Omega_t$ for all $u \in [0, 2]$ and $\alpha \in [3, 4]$. Similar problems have been studied in Li et al. (2024b); Gao et al. (2021); Cho et al. (2024) while the majority of the literature considers solving parameterized PDEs but with either fixed coefficients or fixed domain and initial/boundary conditions. We slightly generalize the problem to consider systems with varying parameters, and with potentially varying domains and initial/boundary conditions.

### 3.2 B-Splines with Basis Functions

In this section, we introduce B-spline basis functions. We begin with the one-dimensional variable $x \in \mathbb{R}$. The B-spline basis functions are defined recursively by the Cox–de Boor formula (Piegl & Tiller, 2012):

$$B_{i,d}(x) = \frac{x - \hat{x}_i}{\hat{x}_{i+d} - \hat{x}_i} B_{i,d-1}(x) + \frac{\hat{x}_{i+d+1} - x}{\hat{x}_{i+d+1} - \hat{x}_{i+1}} B_{i+1,d-1}(x), \tag{4}$$

with the base case

$$B_{i,0}(x) = \begin{cases} 1, & \hat{x}_i \leq x < \hat{x}_{i+1}, \\ 0, & \text{otherwise.} \end{cases} \tag{5}$$

Here, $B_{i,d}(x)$ denotes the value of the $i$-th B-spline basis function of order $d$ evaluated at $x$. The sequence $(\hat{x}_i)_{i=1}^{\ell+d+1}$ is a non-decreasing vector of *knot points*, where $\ell$ is the number of B-spline basis functions. Since a B-spline is a piecewise polynomial function, the knot points determine the interval in which the polynomial is active.

---

[1]Such domain configuration is widely considered in the literature (Takamoto et al., 2022; Gupta & Brandstetter, 2022; Li et al., 2020; Raissi et al., 2019; Wang et al., 2021b; Zhu et al., 2024). Generalizations are considered in Section 5.3 and Section C.2.

There are multiple ways to choose knot points. In this work, we adopt *clamped knot vectors*, where the first and last knots are repeated $d+1$ times, i.e., $\hat{x}_1 = \cdots = \hat{x}_{d+1}$ and $\hat{x}_{\ell+1} = \cdots = \hat{x}_{\ell+d+1}$, with the interior knots equally spaced. For example, on the interval $[0, 3]$ with $\ell = 6$ control points and order $d = 3$, the knot vector is

$$\hat{x} = [0, 0, 0, 0, 1, 2, 3, 3, 3, 3],$$

giving a total of $\ell + d + 1 = 10$ knots.

The B-spline basis functions vector is defined as

$$B_d(x) := [B_{1,d}(x), B_{2,d}(x), \ldots, B_{\ell,d}(x)]^\top, \tag{6}$$

and the coefficients for these basis functions, namely control points, are defined as $c := [c_1, c_2, \ldots, c_\ell]$. Then, we can approximate a solution $s(x)$ with $\hat{s}(x) = cB_d(x)$. Note that with our choice of knot points, we ensure the initial and final values of $\hat{s}(x)$ coincide with the initial and final control points $c_1$ and $c_\ell$. This property will be used later to directly impose initial conditions and Dirichlet boundary conditions with PI-BSNet. A visualization of B-spline basis functions and reconstructions can be found in Fig. 8 in the Appendix.

More generally, for a $n$-dimensional space $x = [x_1, \cdots, x_n] \in \mathbb{R}^n$, we can generate B-spline basis functions based on the Cox-de Boor recursion formula along each dimension $x_i$ with order $d_i$ for $i = 1, 2, \cdots, n$, and the $n$-dimensional control point tensor will be given by $C = [c_{i_1, i_2, \cdots, i_n}]_{\ell_1 \times \ell_2 \times \cdots \times \ell_n}$, where $i_k$ and $\ell_k$ are the index and the number of control points along the $k$-th dimension. We can then approximate the $n$-dimensional surface with B-splines and control points via

$$\hat{s}(x_1, x_2, \cdots, x_n) = \sum_{i_1=1}^{\ell_1} \cdots \sum_{i_n=1}^{\ell_n} c_{i_1, i_2, \cdots, i_n} B_{i_1, d_1}(x_1) \cdots B_{i_n, d_n}(x_n). \tag{7}$$

### 3.3 Physics-Informed B-Spline Networks

In this section, we introduce our proposed physics-informed deep B-spline networks (PI-BSNet). The overall diagram of the network is shown in Fig. 1. The network composites a coefficient network that learns the control point tensor $C$ with system parameters $u$ and ICBC parameters $\alpha$, and the B-spline basis functions $B_{d_i}$ of order $d_i$ for $i = 1, \cdots, n$. We use $G_{\boldsymbol{\theta}}(u, \alpha)(x)$ to denote the PI-BSNet parameterized by $\boldsymbol{\theta}$, where $(u, \alpha)$ is the input to the coefficient net, and $x$ is the input to the B-spline basis. We use $\tilde{C} := G_{\boldsymbol{\theta}}(u, \alpha)$ to denote the control points output by the coefficient network and $\tilde{s}(x) := G_{\boldsymbol{\theta}}(u, \alpha)(x)$ to denote the PI-BSNet prediction. During the forward pass, the control point tensor $\tilde{C}$ output from the coefficient net is multiplied with the B-spline basis functions $B_{d_i}$ via equation 7 to get the approximation $\tilde{s}$. For the backward pass, two losses are imposed to efficiently and effectively train PI-BSNet. We first impose a physics model loss

$$\mathcal{L}_p = \sum_{x \in \mathcal{P}} \frac{1}{|\mathcal{P}|} |\mathcal{F}(\tilde{s}, x, u)|^2, \tag{8}$$

where $\mathcal{F}$ is the governing physics model of the system as defined in equation 1, and $\mathcal{P}$ is the set of points sampled to evaluate the governing physics model. When data is available, we can additionally impose a data loss

$$\mathcal{L}_d = \frac{1}{|\mathcal{D}|} \sum_{x \in \mathcal{D}} |s(x) - \tilde{s}(x)|^2, \tag{9}$$

to capture the mean square error of the approximation, where $s$ is the data point for the high dimensional surface, $\mathcal{D}$ is the set of points where data are available, and $\tilde{s}$ is the prediction from the PI-BSNet. Only for special types of boundary conditions that involve derivatives of the solution (e.g., Neumann and Robin types), we impose the following ICBC loss

$$\mathcal{L}_b = \sum_{x \in \mathcal{M}} \frac{1}{|\mathcal{M}|} |\mathcal{B}(\tilde{s}, x, u)|^2, \tag{10}$$

where $\mathcal{M}$ is the set of points sampled to evaluate the ICBC residual. The total loss is given by

$$\mathcal{L} = w_p \mathcal{L}_p + w_d \mathcal{L}_d + w_b \mathcal{L}_b, \tag{11}$$

where $w_p$, $w_d$ and $w_b$ are the weights for physics, data and ICBC losses, and are usually set to values close to 1.[2] Detailed procedures for training PI-BSNet is shown in Alg. 1 in the Appendix. The choice of the loss function in equation 11 follows the standard formulation used in physics-informed neural networks (Raissi et al., 2019; Karniadakis et al., 2021). Detailed discussions on the effects, failure modes, and alternative designs of such loss functions can be found in (Wang et al., 2021a; Krishnapriyan et al., 2021).

Note that several good properties of B-splines are leveraged in PI-BSNet.

**First, advantageous training efficiency can be obtained with the B-spline representation.** Specifically, the B-spline basis functions are fixed and can be calculated in advance, and the coefficient network is trained to learn only the fixed number of B-spline weights instead of the solution values on the entire space. This speeds up training over existing methods for certain problems.

**Besides, any Dirichlet boundary conditions and initial conditions can be directly assigned via the control points tensor without any learning involved.** This is a natural property of the B-spline representation with clamped knot points (Ahlberg et al., 2016). This feature greatly enhances the accuracy of the learned solution near the ICBC, and improves the ease of design for the loss function as weight factors are often used to impose stronger ICBC constraints in previous literature (Wang et al., 2022a).

**Lastly, the derivatives of the B-spline functions can be analytically calculated.** Specifically, the $p$-th derivative of the $d$-th ordered B-spline at arbitrary point $x$ is given by (Butterfield, 1976)

$$\frac{d^p}{dx^p} B_{i,d}(x) = \frac{(d-1)!}{(d-p-1)!} \sum_{k=0}^{p} (-1)^k \binom{p}{k} \frac{B_{i+k,d-p}(x)}{\prod_{j=0}^{p-1} (\hat{x}_{i+d-j-1} - \hat{x}_{i+k})}. \tag{12}$$

Given this, we can directly calculate derivatives for the back-propagation of physics model loss $\mathcal{L}_p$, which improves both computation efficiency and accuracy over numerical methods.

## 4 Theoretical Analysis

In this section, we provide theoretical guarantees for PI-BSNet. Despite integrating a structured B-spline representation with physics-informed learning, we show that the proposed architecture remains a universal approximator for families of parametric PDEs under mild regularity assumptions. Furthermore, we establish generalization error bounds for learning solution operators associated with elliptic and parabolic PDEs. To the best of our knowledge, such generalization guarantees for parametric PDE families have not been previously studied. All theorem proofs are provided in Appendix A.

### 4.1 Universal Approximation

In this section, we show that the proposed PI-BSNet is a universal approximator for solutions of families of PDEs at arbitrary dimension. We consider $n$ Hilbert spaces $L_2([a_i, b_i])$ for $i = 1, 2, \cdots, n$.

**Assumption 4.1.** The solution of the physics problem defined in equation 1 is continuous in $\alpha$ and $u$. Specifically, let $s_1$ and $s_2$ be the solutions of the physics problem with parameters $\alpha_1, u_1$ and $\alpha_2, u_2$. For any $\epsilon > 0$, there exist $\delta_1 > 0$ and $\delta_2 > 0$ such that given $\|\alpha_1 - \alpha_2\|_2 < \delta_1$, and $\|u_1 - u_2\|_2 < \delta_2$, we have $\|s_1 - s_2\|_2 < \epsilon$.[3]

**Assumption 4.2.** The solution of the physics problem defined in equation 1 is differentiable in $x$.

Assumption 4.1 is a basic assumption for a neural network to approximate solutions of families of parameterized PDEs, and is not strict as it holds for many PDE problems.[4] Assumption 4.2 holds for many PDE problems (Chen et al., 2018; De Angelis, 2015; Barles et al., 2010), and our theoretical results can be generalized to cases where the solution is not differentiable at finite number of points.

---

[2] Ablation experiments on the effects of weights for physics and data losses can be found in Appendix E.5.

[3] Under necessary domain mapping when $\alpha_1 \neq \alpha_2$.

[4] For a well-posed and stable PDE system with unique solution (e.g., linear Poisson, convection-diffusion and heat equations with appropriate ICBCs), change of the system parameter $u$ or the ICBC parameter $\alpha$ usually results in slight change of the value of the solution (Treves, 1962).

**Theorem 4.3.** *Assume Assumption 4.1 and 4.2 hold. For any $n \in \mathbb{N}^+$ dimension, any $u$ and $\alpha$ in a finite parameter set, let $d_i$ be the order of B-spline basis for dimension $i = 1, 2, \cdots, n$. Then for any $d$-time differentiable function $s(x_1, x_2, \cdots, x_n) \in L_2([a_1, b_1] \times [a_2, b_2] \times \cdots \times [a_n, b_n])$ with $d \geq \max\{d_1, \cdots, d_n\}$ where the domain depends on $\alpha$ and the function depends on $u$, and any $\epsilon > 0$, there exist a PI-BSNet configuration $G_{\boldsymbol{\theta}}$ with enough width and depth, and corresponding parameters $\boldsymbol{\theta}^*$ independent of $u$ and $\alpha$ such that*

$$\|\tilde{s} - s\|_2 \leq \epsilon, \tag{13}$$

*where $\tilde{s} = G_{\boldsymbol{\theta}^*}(u, \alpha)(x)$ is the B-spline approximation defined in equation 7 with the control points tensor $G_{\boldsymbol{\theta}^*}(u, \alpha)$.*

Theorem 4.3 tells us that the proposed PI-BSNet is a universal approximator of arbitrary-dimensional surfaces with *varying parameters and domains*. Thus we know that when the solution of the problem defined in equation 1 is unique, and the physics-informed loss functions $\mathcal{L}_p$ is densely imposed and attains zero (De Ryck & Mishra, 2022a; Mishra & Molinaro, 2023), we learn the solution of the PDE problem of arbitrary dimensions. The proof integrates individual universal approximation properties of neural networks (Hornik et al., 1989; Leshno et al., 1993) and B-splines (Blu & Unser, 1999) for the proposed framework.

## 4.2 Generalization Error Bounds

In this section, we provide generalization error bounds of the proposed PI-BSNet for families of elliptic or parabolic PDEs. We make the following assumption about the Lipschitzness of the coefficient network and the training scheme of PI-BSNet.

**Assumption 4.4.** In the PI-BSNet framework, the output of the coefficient network is Lipschitz with respect to its inputs. Specifically, given the coefficient network $G_\theta(u, \alpha)$, $\forall u_1, u_2$ such that $\|u_1 - u_2\|_2 \leq \delta_u$, and $\forall \alpha_1, \alpha_2$ such that $\|\alpha_1 - \alpha_2\|_2 \leq \delta_\alpha$, we have $\|G_\theta(u_1, \alpha_1) - G_\theta(u_1, \alpha_2)\|_2 \leq L(\delta_u + \delta_\alpha)$, for some constant $L$.

**Assumption 4.5.** The training of PI-BSNet is on a finite subset of $\mathcal{U}_{\text{train}} \in \mathcal{U}$ and $\mathcal{A}_{\text{train}} \in \mathcal{A}$ for $u$ and $\alpha$, respectively. The maximum interval between the samples in $\mathcal{U}_{\text{train}}$ and $\mathcal{A}_{\text{train}}$ is $\Delta u$ and $\Delta \alpha$, and $\mathcal{U}_{\text{train}}$, $\mathcal{A}_{\text{train}}$ each fully covers $\mathcal{U}$ and $\mathcal{A}$, i.e., $\forall u_1 \in \mathcal{U}$ and $\alpha_1 \in \mathcal{A}$, there exists $u_2 \in \mathcal{U}_{\text{train}}$ and $\alpha_2 \in \mathcal{A}_{\text{train}}$ such that $\|u_1 - u_2\|_2 \leq \Delta u$, $\|\alpha_1 - \alpha_2\|_2 \leq \Delta \alpha$.

Assumption 4.4 holds in practice as neural networks are usually finite compositions of Lipschitz functions, and its Lipschitz constant can be estimated efficiently (Fazlyab et al., 2019). Assumption 4.5 can be easily achieved since one can sample PDE parameters $u$ and $\alpha$ with equispaced intervals less than $\Delta u$ and $\Delta \alpha$ in $\mathcal{U}$ and $\mathcal{A}$ for training. We then have the following theorem to bound the generalization error for PI-BSNet on the family of elliptic or parabolic PDEs.

**Theorem 4.6.** *Assume Assumption 4.1, Assumption 4.4 and Assumption 4.5 hold. For any elliptic or parabolic PDE with varying parameters $u \in \mathcal{U}$ and $\alpha \in \mathcal{A}$ with $\mathcal{U}$ and $\mathcal{A}$ bounded, suppose that the domain of the PDE $\Omega(\alpha) \in \mathbb{R}^n$ is bounded, $s_{u,\alpha} \in C^0(\bar{\Omega}(\alpha)) \cap C^2(\Omega(\alpha))$ is the solution where $\bar{\Omega}$ is the closure of $\Omega$, $\mathcal{F}(s_{u,\alpha}, x) = 0, x \in \Omega(\alpha)$ defines the PDE, and $\mathcal{B}(s_{u,\alpha}, x) = 0, x \in \Omega_b(\alpha)$ is the boundary condition. Let $G_\theta$ denote a PI-BSNet parameterized by $\theta$ and $\tilde{s}_{u,\alpha} = G_\theta(u, \alpha)$ the solution predicted by PI-BSNet. Let $\text{Unif}(\cdot)$ denote the uniform distribution. If the following conditions holds:*

1. $\mathbb{E}_{x \sim \text{Unif}(\Omega_b(\alpha))}[|\mathcal{B}(G_\theta(u, \alpha), x)|] < \delta_1$, *for all $u \in \mathcal{U}_{train}$ and $\alpha \in \mathcal{A}_{train}$.*

2. $\mathbb{E}_{x \sim \text{Unif}(\Omega(\alpha))}[|\mathcal{F}(G_\theta(u, \alpha), x)|] < \delta_2$, *for all $u \in \mathcal{U}_{train}$ and $\alpha \in \mathcal{A}_{train}$.*

3. $G_\theta(u, \alpha)$, $\mathcal{F}(G_\theta(u, \alpha), \cdot)$, $s(u, \alpha)$ *are $\frac{l}{2}$ Lipschitz continuous on $\Omega(\alpha)$, for all $u \in \mathcal{U}$ and $\alpha \in \mathcal{A}$.*

*Then for any $u \in \mathcal{U}$ and $\alpha \in \mathcal{A}$, the prediction error of $\tilde{s}_{u,\alpha}$ over $\Omega(\alpha)$ is bounded by*

$$\sup_{x \in \Omega(\alpha)} |\tilde{s}_{u,\alpha}(x) - s_{u,\alpha}(x)| \leq \tilde{\delta}_1 + M\tilde{\delta}_2 + \tilde{L}(\Delta u + \Delta \alpha), \tag{14}$$

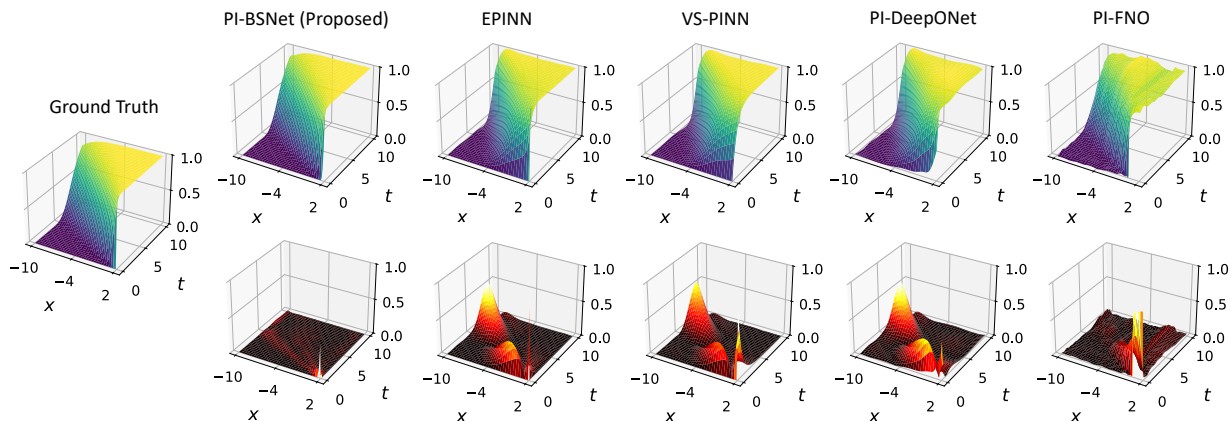

Figure 2: Recovery probability visualizations. Predictions in first row and errors in second row.

*where $M$ is a constant depending on parameter sets $\mathcal{A}$, $\mathcal{U}$, domain functions $\Omega$, $\Omega_b$, and the PDE $\mathcal{F}$, $\tilde{L}$ is some Lipschitz constant, and*

$$\tilde{\delta}_1 = \max_{\alpha} \left\{ \frac{2\delta_1|\Omega_b(\alpha)|}{R_{\Omega_b(\alpha)}|\Omega_b(\alpha)|}, 2l \cdot \left( \frac{\delta_1|\Omega_b(\alpha)| \cdot \Gamma(\frac{n+1}{2})}{lR_{\Omega_b(\alpha)} \cdot \pi^{(n-1)/2}} \right)^{\frac{1}{n}} \right\},$$

$$\tilde{\delta}_2 = \max_{\alpha} \left\{ \frac{2\delta_2|\Omega(\alpha)|}{R_{\Omega(\alpha)}|\Omega(\alpha)|}, 2l \cdot \left( \frac{\delta_2|\Omega(\alpha)| \cdot \Gamma(n/2+1)}{lR_{\Omega(\alpha)} \cdot \pi^{n/2}} \right)^{\frac{1}{n+1}} \right\}, \tag{15}$$

*where $R_{(\cdot)}$ is the regularity of $(\cdot)$, $|(\cdot)|$ denotes the Lebesgue measure and $\Gamma$ is the Gamma function.*

Theorem 4.6 shows that PI-BSNet can generalize reliably across families of elliptic and parabolic PDEs, rather than only matching the training instances. It establishes that small physics and boundary residuals on a well-covered set of training parameters are sufficient to guarantee uniformly small prediction errors for *unseen parameter values*. The result makes explicit how generalization depends on the training coverage of the parameter space and on the smoothness of the learned mapping, thereby offering a principled explanation for generalization across varying PDE parameters and boundary conditions without retraining. Moreover, the proof technique is not specific to the PI-BSNet architecture and can potentially be extended to other network architectures, for which comparable generalization error bounds have not yet been systematically studied.

## 5 Experiments

In this section, we present simulation results on estimating the recovery probability of a dynamical system which gives irregular ICBCs, and compare the proposed PI-BSNet with several baseline methods to show advantages. We then present results to show that PI-BSNet can handle nonhomogeneous ICBCs and learn PDEs on non-rectangular trapezoid domains. All experiment details, ablation experiments and additional experiments can be found in Appendix D, E and F, respectively.

### 5.1 Recovery Probabilities

We consider an autonomous system with dynamics $dx_t = u\,dt + dw_t$, where $x \in \mathbb{R}$ is the state, $w_t \in \mathbb{R}$ is the standard Wiener process with $w_0 = 0$, and $u \in \mathbb{R}$ is the system parameter. Given a set $\mathcal{C}_\alpha = \{x \in \mathbb{R} : x \geq \alpha\}$, we want to estimate the probability of reaching $\mathcal{C}_\alpha$ at least once within time horizon $t$ starting at some $x_0$. Here, $\alpha$ is the varying parameter of the set $\mathcal{C}_\alpha$. Mathematically this can be written as

$$s(x_0, t) := \mathbb{P}\left(\exists \tau \in [0, t], \text{ s.t. } x_\tau \in \mathcal{C}_\alpha \mid x_0\right). \tag{16}$$

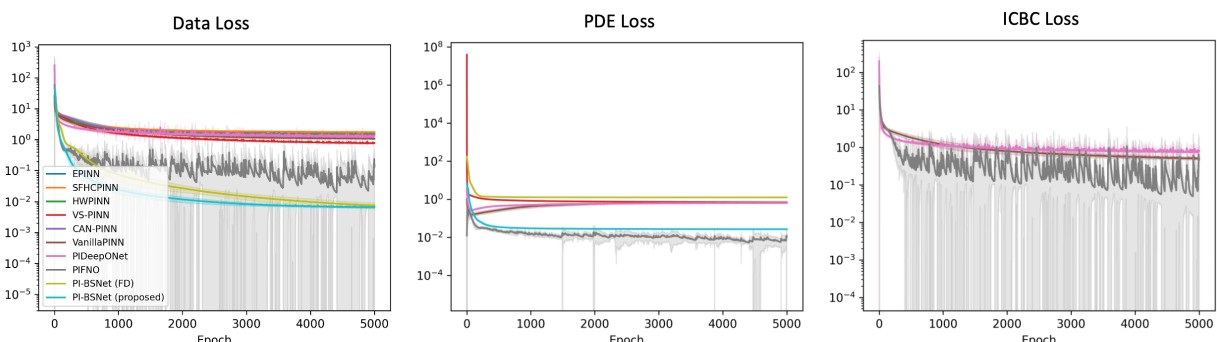

Figure 3: Losses vs. epochs with mean and standard deviation over 10 independent runs.

From Chern et al. (2021) we know that such probability is the solution of convection-diffusion equations with certain initial and boundary conditions

$$\textbf{PDE:} \quad \frac{\partial s}{\partial t}(x,t) - u\frac{\partial s}{\partial x}(x,t) - \frac{1}{2}\left(\frac{\partial^2 s}{\partial x^2}(x,t)\right) = 0, \ \forall [x,t] \in \mathcal{C}_\alpha^c \times \mathcal{T} \tag{17}$$

$$\textbf{ICBC:} \quad s(\alpha,t) = 1, \forall t \in \mathcal{T}, \quad s(x,0) = 0, \forall x \in \mathcal{C}_\alpha^c, \tag{18}$$

where $\mathcal{C}_\alpha^c$ is the complement of $\mathcal{C}_\alpha$, and $\mathcal{T} = [0,T]$ with $T = 10$ be the time horizon of interest. Note that the initial condition and boundary condition at $(x,t) = (\alpha,0)$ is not continuous,[5] which imposes difficulty for learning the solutions.

We train PI-BSNet with 3-layer fully connected neural networks with ReLU activation on varying parameters $u \in [0,2]$ and $\alpha \in [0,4]$ with both data and physics losses,[6] and test on randomly selected parameters in the same domain. We compare PI-BSNet with the standard physics-informed neural network (PINN) (Raissi et al., 2019), PINNs that enforces hard constraints for ICBCs including EPINN (Wang et al., 2023), SFHCPINN (Li et al., 2024a), HWPINN (Chen et al., 2023), efficient PINN methods including VS-PINN (Ko & Park, 2025), CAN-PINN (Chiu et al., 2022), physics-informed neural operator methods including PI-DeepONet (Goswami et al., 2023), PI-FNO (Li et al., 2024b), with similar or comparable NN configurations. All methods are implemented to take PDE and ICBC parameters as additional inputs for parametric PDEs. All comparison experiments are run on a Linux machine with Intel i7 CPU and Nvidia GeForce RTX 4090 GPU. Details of the experiment configuration can be found in Appendix D.

Fig. 2 visualizes the prediction results on a parameter set $(u, \alpha)$ from the same distribution but not used for training (see Fig. 12 in Appendix for full visualization results). We can see that PI-BSNet predicts the ground truth solution accurately during testing while the other methods fails to do so. Fig. 3 visualizes the losses vs. epochs, and we can see that the loss for PI-BSNet drops the fastest and reaches convergence in the shortest amount of time. Fig. 4 visualizes the averaged computation time vs. prediction accuracy over 10 independent runs, and we can see that PI-BSNet obtained the lowest prediction error as well as the lowest training time. This is because PI-BSNet has more compact representation with B-spline basis functions, achieves zero initial and boundary condition losses at the very beginning of the training. In addition, thanks to the analytical calculation of gradients and Hessians, the training time of PI-BSNet is further shortened compared to using finite difference, which is

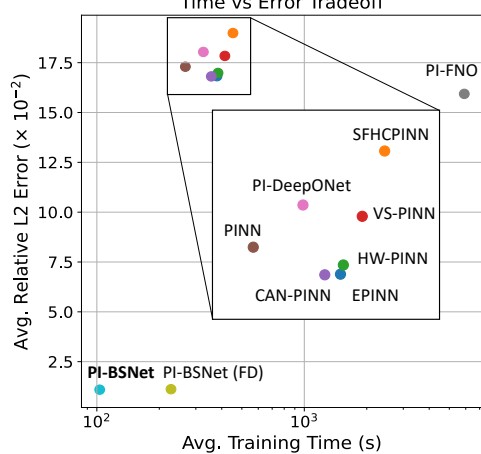

Figure 4: Training time vs. prediction error trade-offs.

[5]When on the boundary of the $\mathcal{C}_\alpha$, the recovery probability at horizon $t = 0$ is $s(\alpha,0) = 1$, but close to the boundary with very small $t$ the recovery probability is $s(x,0) = 0$.
[6]With data, the learning process goes to unique solutions despite the presence of only one BC.

---

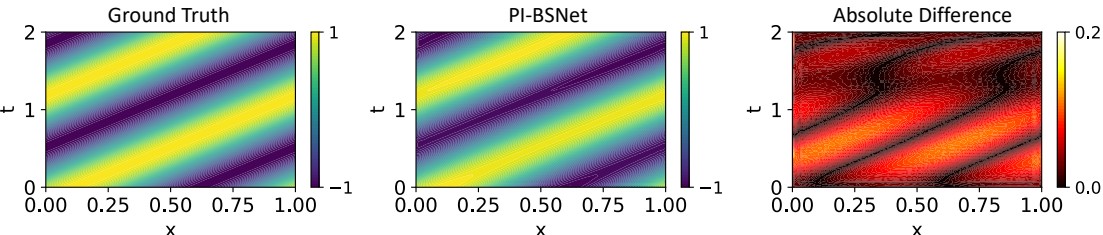

Figure 5: Test results on advection equations with unseen parameter.

given by PI-BSNet (FD). Additional ablations on the effect of direct ICBC assignment can be found in Appendix E.3.

## 5.2 Advection Equations with Nonhomogeneous ICBCs

In this section, we demonstrate the capability of the proposed PI-BSNet to handle nonhomogeneous ICBCs. We consider the following advection equation

$$\frac{\partial s}{\partial t} + u\frac{\partial s}{\partial x} = 0, \tag{19}$$

where $u \in [0.5, 1.5]$ is a changing parameter. The domain of interest is set to be $(x, t) \in [0, 1] \times [0, 2]$, and the initial condition is given by

$$s(x, 0) = A\sin(kx + \alpha), \tag{20}$$

where $A = 1$, $k = 2\pi$, and $\alpha \in [0, 2\pi)$ is a changing parameter. We train PI-BSNet with 3-layer fully connected neural networks with ReLU activation on varying parameters $u \in [0.5, 1.5]$ and $\alpha \in [0, 2\pi)$, and test on randomly selected parameters in the same domain. The B-spline basis of order 5 is used and the number of control points along $x$ and $t$ are set to be $\ell_x = \ell_t = 150$. Note that more control points are used in this case study to represent the high frequency solution. Fig. 5 visualizes the prediction results. The average relative $L_2$ error across 30 test cases is $1.444 \pm 1.352 \times 10^{-1}$. Additional visualizations for the parametric results are shown in Fig. 13 in the Appendix.

## 5.3 Diffusion on Trapezoid

In this experiment, we demonstrate the capability of the proposed PI-BSNet to handle non-rectangular domains. Specifically, we aim to estimate the probability that a driftless Brownian motion in 2D ($x$-$y$ plane) with varying diffusion factor $\alpha \in [0, 1.5]$ along $y$ direction, starting at a point in a trapezoid

$$\Omega_{\text{target}} = \{(x, y) \in \mathbb{R}^2 : y \in [0, 1], \, x \in [-1 + 0.5\,y, \, 1 - 0.5\,y]\}, \tag{21}$$

will exit the domain within a given time horizon $t \in [0, T]$ with $T = 1$. Equivalently, we want to compute the following value for all starting positions $(x, y) \in \Omega_{\text{target}}$ and $t \in [0, T]$

$$s(x, y, t) = \mathbb{P}\Big(\exists \tau \in [0, t] \text{ s.t. } x_\tau \notin \Omega \,\Big|\, (x_0, y_0) = (x, y) \in \Omega_{\text{target}}\Big). \tag{22}$$

We know that the exit probability $s(x, y, t)$ is the solution of the following diffusion equation

$$\frac{\partial s}{\partial t} = \frac{1}{2}\Big(\frac{\partial^2 s}{\partial x^2} + \alpha\frac{\partial^2 s}{\partial y^2}\Big), \tag{23}$$

with ICBCs

$$
\begin{aligned}
s(0, x, y) &= 0, \quad \forall(x, y) \in \Omega_{\text{target}}, \\
s(t, x, y) &= 1, \quad \forall t \in [0, T], \, \forall(x, y) \in \partial\Omega_{\text{target}}.
\end{aligned}
\tag{24}
$$

To solve this problem, we transform the target domain $\Omega_{\text{target}}$ to a rectangular mapped domain $\Omega_{\text{mapped}} = \{(u, v) \in \mathbb{R}^2 : u \in [0, 1], \, v \in [0, 1]\}$, and find the corresponding PDE. We train a PI-BSNet with order $d = 3$,

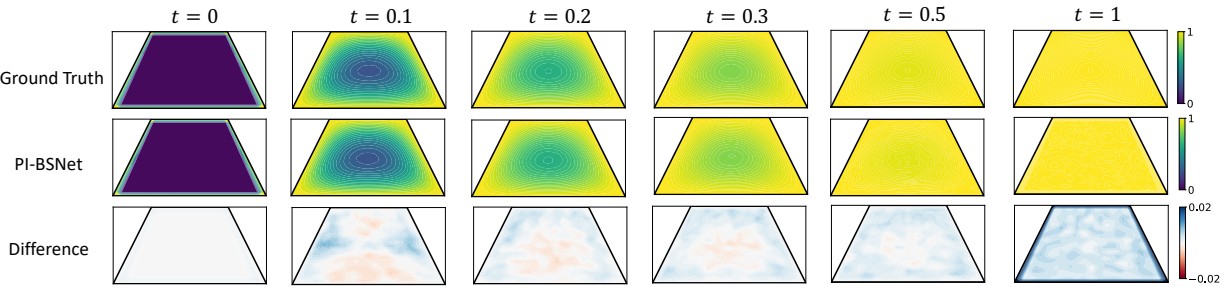

Figure 6: Results on diffusion equation on the trapezoid over time.

number of control points $\ell_x = \ell_y = 20$ and $\ell_t = 100$ on 10 uniformly sampled $\alpha \in [0, 1.5]$. We then test the prediction results on unseen $\alpha$. Fig. 6 visualizes the results for one test case. It can be seen that PI-BSNet can accurately predict the diffusion evolution on the trapezoid with unseen parameter. The average prediction relative $L_2$ error is $3.172 \pm 1.580 \times 10^{-3}$, over 10 random testing trials. Domain transformation derivations and experiment details can be found in Appendix D.3. Additional visualizations for the parametric results are shown in Fig. 14 in the Appendix.

## 6    Discussions

In this section, we discuss the practical considerations and limitations.

**Choice of control point numbers.** Theorem 4.3 suggests that a higher number of control points can provide a lower approximation error. In practice, when data are available, which is common for parametric PDE learning problems, one could tune the number of control points in prior to training to reach desired approximation error, or choose higher numbers of control points to ensure expressiveness of the model. In general, smoother problems will require less control points to reach a certain desired error tolerance, since B-splines are smooth functions. The number of control points has a direct impact on the training efficiency of PI-BSNet. Specifically, for a $n$-dimensional problem, the number of neurons in the last layer of the coefficient network scales $O(\ell^n)$ with $\ell$ being the number of control points along each dimension. Empirically, the number of NN parameters and the training time both increase as the number of control points increases, as shown in Table 4 in Appendix E.4, where we ablate different number of control points. In contrast, the test-time forward pass of the model remains computationally inexpensive and exhibits negligible dependence on the number of control points. In our implementation, the forward pass time is on the order of $10^{-4}$ seconds and is largely dominated by framework-level overhead (e.g., tensor allocation and dispatch in PyTorch), rather than the spline evaluation or neural network computation itself.

**Scalability to high-dimensional systems.** For the number of points used to enforce the PDE loss, the proposed PI-BSNet, along with many PINN and neural operator methods, suffers from the curse of dimensionality. However, adaptive sampling techniques, such as those discussed in (Zeng et al., 2022), can be effective in reducing the number of points required for training PINNs, especially when the PDE solution is concentrated in certain regions of the domain. Similar ideas can be potentially incorporated into the proposed PI-BSNet framework through adaptive knot placement (Yeh et al., 2020), which would reduce the number of control points needed and improve scalability.

For the number of network parameters, using fixed B-spline basis will result in exponential increase with the dimension for the proposed PI-BSNet. Additional treatment such as hierarchical B-splines (Valentin, 2019), control point adjustments (Yeh et al., 2020; Yang et al., 2004) and non-uniform B-spline representations (Piegl & Tiller, 2012) can be potentially used to reduce the number of control points, thus effectively reduce the network size. Nevertheless, the proposed PI-BSNet requires less control points than typical grid-based optimization methods for PDE solving in such cases, thanks to the representation capacity of B-spline basis functions.

**General domains and transformations.** While the standard B-spline formulation primarily supports domains that are diffeomorphic to rectangulars, we show in Appendix C.2 that B-splines can effectively represent solutions on general domains, which motivates future work on physics-informed learning for such cases. In addition, coordinate transformations such as Mojgani et al. (2023) can be potentially leveraged to effectively reduce the representation complexity, for advection-dominated equations such as equation 19, where dense control points are typically needed to accurately capture the coupled evolution between state and time.

**Fixed basis vs. learned basis.** With the use of fixed B-spline basis, the proposed method enjoys unique features such as exact enforcement of initial and boundary conditions, analytical derivative calculations for physics loss functions, inherent solution smoothness, and a compact, structured representation. These lead to practical advantages for systems like the one studied in Section 5.1, where we demonstrate faster convergence, higher prediction accuracy, and reduced computation time. In the literature, there are also methods such as PI-DeepONet (Goswami et al., 2023) that leverage learned basis. Correspondingly, longer training time is usually required due to the lack of structural advantages, with a potential gain in expressiveness. This distinction reflects the difference of architecture choice results in different capabilities and target applications. In particular, our method is especially suited for scenarios that demand strict IC/BC satisfaction, smooth solutions, and efficient training.

## 7 Conclusion

In this paper, we propose physics-informed deep B-spline networks (PI-BSNet), which incorporate B-spline functions into physics-informed neural networks, to efficiently learn solutions of families of PDEs with varying ICBCs. With PI-BSNet, analytical derivatives are available for B-splines to calculate physics-informed losses, initial conditions and Dirichlet boundary conditions can be directly imposed through B-spline control points. We prove theoretical guarantees that PI-BSNets are universal approximators and have bounded generalization errors for elliptic and parabolic PDE families. We demonstrate in experiments that PI-BSNet achieves better training time and prediction accuracy trade-offs over various baselines, and is capable of addressing nonhomogeneous ICBCs and non-rectangular domains. Future work includes extensions to unparameterized domains and high-dimensional problems. Besides, the structured and smooth spline representation underlying PI-BSNet provides a natural pathway to further extend the proposed framework, as fundamental B-spline properties such as convex hull guarantees, inherent smoothness, and noise-filtering behavior can support uncertainty quantification, robustness analysis, and other promising future directions toward reliable and interpretable physics-informed learning (Prautzsch, 2002; De Boor & De Boor, 1978; Wahba, 1990).

## Acknowledgments

This material is based upon work supported in part by the National Science Foundation under Grant number 2442948, and in part by a grant from the Commonwealth of Pennsylvania, Department of Community and Economic Development. Saviz Mowlavi is supported solely by Mitsubishi Electric Research Laboratories. We thank Jasmine Ratchford for insightful discussions that helped shape this work. We are also grateful to Giovanni Leoni from the Department of Mathematics at Carnegie Mellon University for valuable discussions on the continuity of PDE solutions.

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

## Table of Contents

# A  Proof of Theorems

## A.1  Universal Approximation (Theorem 4.3)

In this section, we prove that PI-BSNets are universal approximators of families of PDEs of arbitrary dimensions.

We first consider the one-dimensional function space $L_2([a, b])$ with $L_2$ norm defined over the interval $[a, b]$. For two functions $s, g \in L_2([a, b])$, we define the inner product of these two functions as

$$\langle s, g \rangle := \int_a^b s(x) g^*(x) dx, \tag{25}$$

where $*$ denotes the conjugate complex. We say a function $s(x)$ is square-integrable if the following holds

$$\langle s, s \rangle = \int_a^b |s(x)|^2 dx < \infty. \tag{26}$$

We define the $L_2$ norm between two functions $s, g$ as

$$\|s - g\|_2 := \left( \int_a^b |s(x) - g(x)|^2 dx \right)^{\frac{1}{2}}. \tag{27}$$

We then state the following theorem that shows B-spline functions are universal approximators in the sense of $L_2$ norms in one dimension.

**Theorem A.1.** *Given a positive natural number $d$ and any $d$-time differentiable function $s(x) \in L_2([a, b])$, then for any $\epsilon > 0$, there exist a positive natural value $\bar{\ell}$ such that for all $\ell \geq \bar{\ell}$, there exists a realization of control points $c_1, c_2, \cdots, c_\ell$ such that*

$$\|s - \hat{s}\|_2 \leq \epsilon, \tag{28}$$

*where*

$$\hat{s}(x) = \sum_{i=1}^{\ell} c_i B_{i,d}(x)$$

*is the B-spline approximation with $B_{i,d}(x)$ being the B-spline basis functions defined in equation 6.*

*Proof.* (Theorem A.1) See Section B.1. □

Now that we have the error bound of B-spline approximations in one dimension, we will extend the results to arbitrary dimensions. We point out that the space $L_2([a, b])$ is a Hilbert space (Balakrishnan, 2012). We consider $n$ Hilbert spaces $L_2([a_i, b_i])$ for $i = 1, 2, \cdots, n$. We define the inner products of two $n$-dimensional functions $s, g \in L_2([a_1, b_1] \times \cdots \times [a_n, b_n])$ as

$$\langle s, g \rangle := \int_{a_n}^{b_n} \cdots \int_{a_1}^{b_1} s(x_1, \cdots, x_n) g^*(x_1, \cdots, x_n) dx_1 \cdots dx_n, \tag{29}$$

and we say a function $s : \mathbb{R}^n \to \mathbb{R}$ is square-integrable if

$$\langle s, s \rangle = \int_{a_n}^{b_n} \cdots \int_{a_1}^{b_1} |s(x_1, \cdots, x_n)|^2 dx_1 \cdots dx_n < \infty. \tag{30}$$

Now we present the following lemma to bound the approximation error of $n$-dimensional B-splines.

**Lemma A.2.** *Given a set positive natural numbers $d_1, \cdots, d_n$ and a $d$-time differentiable function $s(x_1, x_2, \cdots, x_n) \in L_2([a_1, b_1] \times [a_2, b_2] \times \cdots \times [a_n, b_n])$. Assume $d \geq \max\{d_1, \cdots, d_n\}$, then given any $\epsilon > 0$, there exist $\bar{\ell}_i \in \mathbb{N}^+$ of control points for each component $i = 1, ..., n$, such for all $\ell_i \geq \bar{\ell}_i, \forall i = 1, \cdots, n$, there exists a control points realization such that*

$$\|s - \hat{s}\|_2 \leq \epsilon, \tag{31}$$

*where*

$$\hat{s}(x_1, \cdots, x_n) = \sum_{i_1=1}^{\ell_1} \cdots \sum_{i_n=1}^{\ell_n} c_{i_1, \cdots, i_n} B_{i_1, d_1}(x_1) \cdots B_{i_n, d_n}(x_n). \tag{32}$$

*Proof.* (Lemma A.2) See Section B.2. □

On the other hand, we know that neural networks are universal approximators (Hornik et al., 1989; Leshno et al., 1993), i.e., with large enough width or depth a neural network can approximate any function with arbitrary precision. We first show that given some basic assumptions on the solution of the physics problems, the optimal control points are continuous in the system and domain parameters $u$ and $\alpha$, thus can be approximated by neural networks.

In this following lemma, we show that with Assumption 4.1 and Assumption 4.2, the optimal control points are continuous in terms of the system and ICBC parameters.

**Lemma A.3.** *For any $n \in \mathbb{N}^+$ and two $n$-dimensional surfaces $s_1, s_2 \in L_2([a_1, b_1] \times [a_2, b_2] \times \cdots \times [a_n, b_n])$ being the solution of the physics problem defined in equation 1 with parameters $\alpha_1, u_1$ and $\alpha_2, u_2$. Assume Assumption 4.1 and Assumption 4.2 hold. Let $C_1$ and $C_2$ be the two control points tensors that reconstruct $\hat{s}_1$ and $\hat{s}_2$. For any $\epsilon > 0$, $\epsilon_1, \epsilon_2 > 0$, there exist $\delta_1, \delta_2 > 0$ such that $\|\alpha_1 - \alpha_2\| < \delta_1$, and $\|u_1 - u_2\| < \delta_2$, and control points tensors $C_1$ and $C_2$ with $\|C_1 - C_2\| < \delta(\epsilon)$ such that $\|s_1 - \hat{s}_1\|_2 < \epsilon_1$, $\|s_2 - \hat{s}_2\|_2 < \epsilon_2$, and $\|s_1 - s_2\|_2 < \epsilon$. Here $\delta(\epsilon) \to 0$ when $\epsilon \to 0$.*

*Proof.* (Lemma A.3) See Section B.3. □

Lemma A.3 shows that the optimal control points exist and are continuous in $\alpha$ and $u$, thus can be approximated by neural networks with arbitrary precision given enough representation capability (Hornik et al., 1989). We restate the universal approximation theorem for neural networks in our context as follows, assuming the requirements for the neural network are met. [7]

---

[7]The Borel space assumptions are met since we consider $L_2$ space which is a Borel space.

**Theorem A.4.** *Assume Assumption 4.1 and 4.2 hold. Given any $u$ and $\alpha$ in a finite parameter set, and the corresponding B-spline approximation control points tensor $C := [c]_{\ell_1 \times \cdots \times \ell_n}$, for the coefficient net $G_{\boldsymbol{\theta}}(u, \alpha)$ and $\forall \epsilon > 0$, when the network has enough width and depth, there is $\boldsymbol{\theta}^*$ such that*

$$\|G_{\boldsymbol{\theta}^*}(u, \alpha) - C\| \leq \epsilon. \tag{33}$$

Then, we combine Lemma A.2 and Theorem A.4 to prove the universal approximation property of PI-BSNet (Theorem 4.3).

*Proof.* (Theorem 4.3) For any $u$ and $\alpha$, from Lemma A.2 we know that there is $\ell_1, \cdots, \ell_n$ by taking the upper bounds of control point numbers (by setting $\ell_i \geq \bar{\ell}_i, \forall i = 1, \cdots, n$), and the control points realization $C := [c]_{\ell_1 \times \cdots \times \ell_n}$ such that $\|s(x_1, x_2, \cdots, x_n) - \hat{s}(x_1, x_2, \cdots, x_n)\|_2 \leq \epsilon_1$ for any $\epsilon_1 > 0$, where $\hat{s}$ is the B-spline approximation defined in equation 7 with the control points tensor $C$. Then, from Theorem A.4 we know that there is a DBSN configuration $G_{\boldsymbol{\theta}}(u, \alpha)$ and corresponding parameters $\boldsymbol{\theta}^*$ such that $\|G_{\boldsymbol{\theta}^*}(u, \alpha) - C\| \leq \epsilon_2$ for any $\epsilon_2 > 0$. Since B-spline functions are defined on bounded domains and are continuous and Lipschitz (Prautzsch, 2002; Kunoth et al., 2018), and $\tilde{s}$ and $\hat{s}$ are weighted sum of B-spline basis functions with weights $G_{\boldsymbol{\theta}^*}(u, \alpha)$ and $C$ (see equation 7), we know that $\|\tilde{s} - \hat{s}\|_2 \leq L\epsilon_2$ for some positive constant $L$. Then by triangle inequality of the $L_2$ norm, we have

$$\|\tilde{s} - s\|_2 \leq \|\tilde{s} - \hat{s}\|_2 + \|\hat{s} - s\|_2 \leq \epsilon_1 + L\epsilon_2. \tag{34}$$

For any $\epsilon > 0$ we can find $\epsilon_1$ and $\epsilon_2$ such that $\epsilon = \epsilon_1 + L\epsilon_2$ to bound the norm. $\square$

## A.2 Generalization Error Bounds (Theorem 4.6)

In this section, we show proof of Theorem 4.6 to bound the generalization error of PI-BSNet on families of PDEs.

We start by giving a lemma on PI-BSNet generalization error for fixed elliptic and parabolic PDEs. From Section 4 we know that PI-BSNets are universal approximators of PDEs. From Section 3 we know that the physics loss $\mathcal{L}_p$ is imposed over the domain of interest. Given this, we have the following lemma on the generalization error of PI-BSNet for elliptic and parabolic PDEs with *fixed* parameters, adapted from (Wang & Nakahira, 2023, Theorem 6) and (Peng et al., 2020, Theorem 2.4).

**Definition A.5.** Let $\Omega \subset \mathbb{R}^n$ be a domain. We define the regularity of $\Omega$ as

$$R_\Omega := \inf_{x \in \Omega, r > 0} \frac{|B(x, r) \cap \Omega|}{\min \left\{ |\Omega|, \frac{\pi^{n/2} r^n}{\Gamma(n/2 + 1)} \right\}}, \tag{35}$$

where $B(x, r) := \{y \in \mathbb{R}^n \mid \|y - x\| \leq r\}$ and $|(\cdot)|$ is the Lebesgue measure of a set $(\cdot)$.

**Lemma A.6.** *(Wang et al., 2025, Theorem 5) Assume the PDE is elliptic or parabolic, for any fixed parameters $\alpha$ and $u$, suppose that $\Omega \in \mathbb{R}^n$ is a bounded domain, $s \in C^0(\bar{\Omega}) \cap C^2(\Omega)$ is the solution to the PDE of interest where $\bar{\Omega}$ is the closure of $\Omega$, $\mathcal{F}(s, x) = 0, x \in \Omega$ defines the PDE, and $\mathcal{B}(s, x) = 0, x \in \Omega_b$ is the boundary condition. Let $G_\theta$ denote a PI-BSNet parameterized by $\theta$ and $\tilde{s}$ the solution predicted by PI-BSNet. If the following conditions holds:*

1. $\mathbb{E}_{x \sim \text{Unif}(\Omega_b(\alpha))} [|\mathcal{B}(G_\theta, x)|] < \delta_1$, *where* $\text{Unif}(\Omega_b)$ *is the uniform distribution in $\Omega_b$.*

2. $\mathbb{E}_{x \sim \text{Unif}(\Omega(\alpha))} [|\mathcal{F}(G_\theta, x)|] < \delta_2$, *where* $\text{Unif}(\Omega)$ *is the uniform distribution in $\Omega$.*

3. $G_\theta, \mathcal{F}(G_\theta, \cdot), s$ *are* $\frac{l}{2}$ *Lipschitz continuous on $\Omega$.*

*Then the error of $\tilde{s}$ over $\Omega$ is bounded by*

$$\sup_{x \in \Omega} |\tilde{s}(x) - s(x)| \leq \tilde{\delta}_1 + M\tilde{\delta}_2 \tag{36}$$

*where $M$ is a constant depending on $\Omega$, $\Omega_b$ and $\mathcal{F}$, and*

$$\tilde{\delta}_1 = \max\left\{ \frac{2\delta_1|\Omega_b|}{R_{\Omega_b}|\Omega_b|}, 2l \cdot \left(\frac{\delta_1|\Omega_b| \cdot \Gamma(\frac{n+1}{2})}{lR_{\Omega_b} \cdot \pi^{(n-1)/2}}\right)^{\frac{1}{n}} \right\},$$

$$\tilde{\delta}_2 = \max\left\{ \frac{2\delta_2|\Omega|}{R_{\Omega}|\Omega|}, 2l \cdot \left(\frac{\delta_2|\Omega| \cdot \Gamma(n/2+1)}{lR_{\Omega} \cdot \pi^{n/2}}\right)^{\frac{1}{n+1}} \right\},$$

(37)

*where $R_{(\cdot)}$ is the regularity of $(\cdot)$, $|(\cdot)|$ is the Lebesgue measure of a set $(\cdot)$ and $\Gamma$ is the Gamma function.*

We then prove Theorem 4.6 to bound the generalization error for PI-BSNet on the family of PDEs.

*Proof.* (Theorem 4.6 ) The goal is to prove equation 14 holds for any $u \in \mathcal{U}$ and $\alpha \in \mathcal{A}$. Without loss of generality, we pick arbitrary $u_1 \in \mathcal{U}$ and $\alpha_1 \in \mathcal{A}$ to evaluate the prediction error, and we denote the ground truth and PI-BSNet prediction as $s_1$ and $\tilde{s}_1$, respectively. From Assumption 4.5 we know that there are $u_2 \in \mathcal{U}_{\text{train}}$ and $\alpha_2 \in \mathcal{A}_{\text{train}}$ such that $\|u_1 - u_2\|_2 \leq \Delta u$, $\|\alpha_1 - \alpha_2\|_2 \leq \Delta \alpha$. Let $s_2$ and $\tilde{s}_2$ denote the ground truth and PI-BSNet prediction on the PDE with parameters $u_2$ and $\alpha_2$. Since the conditions in Theorem 4.6 hold for all $u \in \mathcal{U}_{\text{train}}$ and $\alpha \in \mathcal{A}_{\text{train}}$, and $\tilde{\delta}_1$ and $\tilde{\delta}_2$ are taking the maximum among all $\alpha$, we know the following inequality holds due to Lemma A.6.

$$\sup_{x \in \Omega(\alpha_1)} |\tilde{s}_2(x) - s_2(x)| \leq \tilde{\delta}_1 + M\tilde{\delta}_2,$$

(38)

where $M$ is a constant depending on $\Omega(\alpha_2)$, $\Omega_b(\alpha_2)$ and $\mathcal{F}$, and $\tilde{\delta}_1$, $\tilde{\delta}_2$ are given by equation 15. Note that the domain considered is $\Omega(\alpha_1)$, as eventually we will bound the error in this domain. Necessary mapping of the domain is applied here and in the rest of the proof when $\alpha_1 \neq \alpha_2$.

Since $\mathcal{A}$ and $\mathcal{U}$ are bounded, and from Assumption 4.1 we know the PDE solution is continuous in $u$ and $\alpha$, we know the solution is Lipschitz in $u$ and $\alpha$. Then we have

$$\sup_{x \in \Omega(\alpha_1)} |s_1(x) - s_2(x)| = \|s_1 - s_2\|_\infty \leq K_1\|s_1 - s_2\|_2 \leq L_1(\Delta u + \Delta \alpha),$$

(39)

where $K_1$ and $L_1$ are some finite constants (Evans, 2022, Chapter 6 and Chapter 7).

Lastly, from Assumption 4.4 we know that the learned control points from the coefficient network $G_\theta(u, \alpha)$ are Lipschitz in $u$ and $\alpha$. Since the B-spline basis functions $B_{i,d}(x)$ are bounded by construction, we know that

$$\sup_{x \in \Omega(\alpha_1)} |\tilde{s}_1(x) - \tilde{s}_2(x)| = \|\tilde{s}_1 - \tilde{s}_2\|_\infty \leq K_2\|\tilde{s}_1 - \tilde{s}_2\|_2 \leq L_2(\Delta u + \Delta \alpha),$$

(40)

where $K_2$ and $L_2$ are some finite constants.

Now, combining equation 38, equation 39 and equation 40, by triangular inequality we get

$$\begin{aligned}
&\sup_{x \in \Omega(\alpha_1)} |\tilde{s}_1(x) - s_1(x)| \\
&\leq \sup_{x \in \Omega(\alpha_1)} |\tilde{s}_2(x) - s_2(x)| + \sup_{x \in \Omega(\alpha_1)} |s_1(x) - s_2(x)| + \sup_{x \in \Omega(\alpha_1)} |\tilde{s}_1(x) - \tilde{s}_2(x)| \\
&\leq \tilde{\delta}_1 + M\tilde{\delta}_2 + \tilde{L}(\Delta u + \Delta \alpha),
\end{aligned}$$

(41)

where $M$ is a constant depending on parameter sets $\mathcal{A}$, $\mathcal{U}$, domain functions $\Omega$, $\Omega_b$, and the PDE $\mathcal{F}$, $\tilde{L} = L_1 + L_2$ is a Lipschitz constant. $\qquad\square$

# B  Proof of Supporting Theorems and Lemmas

## B.1  Proof of Theorem A.1

*Proof.* (Theorem A.1) From (Jia & Lei, 1993; Strang & Fix, 1971) we know that given $d$ the least square spline approximation of $\hat{s}(x) = \sum_{i=1}^{\ell} c_i B_{i,d}(x)$ can be obtained by applying pre-filtering, sampling and post-filtering

on $s$, with $L_2$ error bounded by

$$\|s - \hat{s}\|_2 \leq C_d \cdot T^d \cdot \|s^{(d)}\|, \tag{42}$$

where $C_d$ is a known constant (Blu & Unser, 1999), $T$ is the sampling interval of the pre-filtered function, and $\|s^{(d)}\|$ is the norm of the $d$-th derivative of $s$ defined by

$$\left\|s^{(d)}\right\| = \left(\frac{1}{2\pi} \int_{-\infty}^{+\infty} \omega^{2d} |S(\omega)|^2 d\omega\right)^{1/2}, \tag{43}$$

and $S(\omega)$ is the Fourier transform of $s(x)$. Note that given $s$ and $d$, $\left\|s^{(d)}\right\|$ is a known constant.

Then, from (Unser, 1999) we know that the samples from the pre-filtered functions are exactly the control points $c_i$ that minimize the $L_2$ norm of the approximation error. In other words, the sampling time $T$ and the number of control points $\ell$ are coupled through the following relationship

$$T = \frac{b - a}{\ell - 1}, \tag{44}$$

since the domain is $[a, b]$ and it is divided into $\ell - 1$ equispaced intervals for control points. Then with $c_i$ being the samples with interval $T$, we can rewrite the error bound into

$$\|s - \hat{s}\|_2 \leq C_d \cdot \left(\frac{b - a}{\ell - 1}\right)^d \cdot \|s^{(d)}\|. \tag{45}$$

Thus we know that for $\forall \epsilon > 0$, we can find $\bar{\ell} \in \mathbb{N}^+$ such that for all $\ell \geq \bar{\ell}$

$$\|s - \hat{s}\|_2 \leq \frac{(b - a)^d C_d \|s^{(d)}\|}{(\ell - 1)^d} \leq \epsilon, \tag{46}$$

because for fixed $d$ the numerator is a constant, and the $L_2$ norm bound converges to 0 as $\ell \to \infty$. $\qquad\square$

## B.2 Proof of Lemma A.2

*Proof.* (Lemma A.2) For given $\ell_1, \cdots, \ell_n$, let $C := [c]_{\ell_1 \times \cdots \times \ell_n}$ be the control points tensor such that $\|s(x_1, x_2, \cdots, x_n) - \hat{s}(x_1, x_2, \cdots, x_n)\|_2$ is minimized. Let $(x_1', x_2', \cdots, x_n')$ denote the knot points in the $n$-dimensional space, i.e., the equispaced grids where the control points are located. Then from Theorem A.1 and the separability of the B-splines (Unser et al., 2002, Section IV-E), we know that

$$\int_{a_1}^{b_1} (s - \hat{s})(s - \hat{s})^*(x_1, x_2', \cdots, x_n') dx_1 \leq \epsilon_{x_1}, \tag{47}$$

where $\epsilon_{x_1} = \frac{(b-a)^{d_1} C_{d_1} \|s^{(d_1)}\|}{(\ell_1 - 1)^{d_1}}$. This shows that the $L_2$ norm along the $x_1$ direction at any knots points $(x_2', \cdots, x_n')$ is bounded. Now we show the following is bounded

$$\int_{a_2}^{b_2} \int_{a_1}^{b_1} (s - \hat{s})(s - \hat{s})^*(x_1, x_2, x_3', \cdots, x_n') dx_1 dx_2. \tag{48}$$

We argue that $s$ is Lipschitz as it is defined on a bounded domain and is $d$-time differentiable, and $\hat{s}$ is also Lipschitz as B-spline functions of any order are Lipschitz (Prautzsch, 2002; Kunoth et al., 2018) and $C$ is finite. Then we know that $(s - \hat{s})(s - \hat{s})^*$ is Lipschitz with some Lipschitz constant $L_{x_i}$ along dimension $i$ for $i = 1, 2, \cdots, n$.

Let $\{x_i^{(j)}\}_{j=1}^{\ell_i}$ denote the equispaced knot points on $[a_i, b_i]$ with spacing $\frac{b_i - a_i}{\ell_i - 1}$. Define a partition $\{I_{i,j}\}_{j=1}^{\ell_i}$ of $[a_i, b_i]$ such that for all $x_i \in I_{i,j}$, $|x_i - x_i^{(j)}| \leq \frac{b_i - a_i}{2(\ell_i - 1)}$. Then for $i = 2$, for all $x_2 \in I_{2,j}$,

$$|(s - \hat{s})(s - \hat{s})^*(x_1, x_2, x_3', \cdots, x_n') - (s - \hat{s})(s - \hat{s})^*(x_1, x_2^{(j)}, x_3', \cdots, x_n')| \leq L_{x_2} \frac{b_2 - a_2}{2(\ell_2 - 1)}. \tag{49}$$

Then we have

$$\int_{a_2}^{b_2} \int_{a_1}^{b_1} (s - \hat{s})(s - \hat{s})^*(x_1, x_2, x_3', \cdots, x_n') dx_1 dx_2 \tag{50}$$

$$= \sum_{j=1}^{\ell_2} \int_{I_{2,j}} \int_{a_1}^{b_1} (s - \hat{s})(s - \hat{s})^*(x_1, x_2, x_3', \cdots, x_n') dx_1 dx_2$$

$$\leq \sum_{j=1}^{\ell_2} \int_{I_{2,j}} \int_{a_1}^{b_1} (s - \hat{s})(s - \hat{s})^*(x_1, x_2^{(j)}, x_3', \cdots, x_n') dx_1 dx_2$$

$$+ \sum_{j=1}^{\ell_2} \int_{I_{2,j}} \int_{a_1}^{b_1} |(s - \hat{s})(s - \hat{s})^*(x_1, x_2, x_3', \cdots, x_n') - (s - \hat{s})(s - \hat{s})^*(x_1, x_2^{(j)}, x_3', \cdots, x_n')| dx_1 dx_2 \tag{51}$$

$$\leq \sum_{j=1}^{\ell_2} \int_{I_{2,j}} \int_{a_1}^{b_1} (s - \hat{s})(s - \hat{s})^*(x_1, x_2^{(j)}, x_3', \cdots, x_n') dx_1 dx_2 + \int_{a_2}^{b_2} \int_{a_1}^{b_1} L_{x_2} \frac{b_2 - a_2}{2(\ell_2 - 1)} dx_1 dx_2 \tag{52}$$

$$\leq (b_2 - a_2) \left[ \epsilon_{x_1} + L_{x_2} \frac{(b_2 - a_2)(b_1 - a_1)}{2(\ell_2 - 1)} \right] := \epsilon_{x_1, x_2}, \tag{53}$$

where equation 51 is the triangle inequality of norms, and equation 52 is due to the Lipschitz-ness of the function.

Similarly we can show the bound when we integrate the next dimension

$$\int_{a_3}^{b_3} \int_{a_2}^{b_2} \int_{a_1}^{b_1} (s - \hat{s})(s - \hat{s})^*(x_1, x_2, x_3, x_4', \cdots, x_n') dx_1 dx_2 dx_3 \tag{54}$$

$$\leq \sum_{j=1}^{\ell_3} \int_{I_{3,j}} \int_{a_2}^{b_2} \int_{a_1}^{b_1} (s - \hat{s})(s - \hat{s})^*(x_1, x_2, x_3^{(j)}, x_4', \cdots, x_n') dx_1 dx_2 dx_3 + \int_{a_3}^{b_3} \int_{a_2}^{b_2} \int_{a_1}^{b_1} L_{x_3} \frac{b_3 - a_3}{2(\ell_3 - 1)} dx_1 dx_2 dx_3 \tag{55}$$

$$\leq (b_3 - a_3) \left[ \epsilon_{x_1, x_2} + L_{x_3} \frac{(b_3 - a_3)(b_2 - a_2)(b_1 - a_1)}{2(\ell_3 - 1)} \right] := \epsilon_{x_1, x_2, x_3}. \tag{56}$$

We know that $\epsilon_{x_1, x_2, x_3} \to 0$ when $\ell_i \to \infty$ for $i = 1, 2, 3$. By keeping doing this, recursively we can find the bound $\epsilon_{x_1, \cdots, x_n}$ that

$$\int_{a_n}^{b_n} \cdots \int_{a_1}^{b_1} (s - \hat{s})(s - \hat{s})^*(x_1, \cdots, x_n) dx_1 \cdots dx_n \leq \epsilon_{x_1, \cdots, x_n}, \tag{57}$$

where the left hand side is exactly $\|s(x_1, x_2, \cdots, x_n) - \hat{s}(x_1, x_2, \cdots, x_n)\|_2^2$, and the right hand side $\epsilon_{x_1, \cdots, x_n} \to 0$ when $\ell_i \to \infty$ for all $i = 1, 2, \cdots, n$. Thus for any $\epsilon > 0$, we can find $\bar{\ell}_i$ for $i = 1, 2, \cdots, n$ such that for all $\ell_i \geq \bar{\ell}_i$, there exists a control points realization such that

$$\|s(x_1, x_2, \cdots, x_n) - \hat{s}(x_1, x_2, \cdots, x_n)\|_2 \leq \epsilon. \tag{58}$$

$\square$

## B.3 Proof of Lemma A.3

*Proof.* (Lemma A.3) From Assumption 4.1 we know that there exists $\delta_1, \delta_2 > 0$ such that $\|\alpha_1 - \alpha_2\| < \delta_1$, and $\|u_1 - u_2\| < \delta_2$, and $\|s_1 - s_2\|_2 < \epsilon$.

Now we need to prove that there exist control points tensors $C_1$ and $C_2$ with $\|C_1 - C_2\| < \delta(\epsilon)$ such that $\|s_1 - \hat{s}_1\|_2 < \epsilon_1$, $\|s_2 - \hat{s}_2\|_2 < \epsilon_2$. We prove by construction.

We first construct surrogate functions $\bar{s}_1$ and $\bar{s}_2$ by interpolation of $s_1$ and $s_2$, then find B-spline approximations $\hat{s}_1$ and $\hat{s}_2$ of the surrogate functions. The relationships between $s$, $\bar{s}$ and $\hat{s}$ are visualized in Fig. 7.

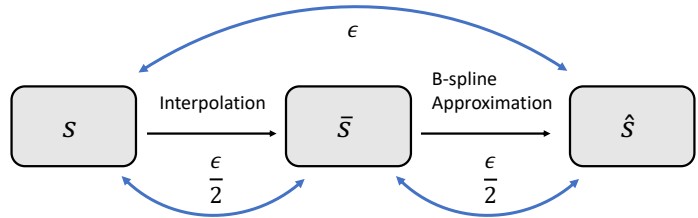

Figure 7: Relationships between ground truth $s$, interpolation $\bar{s}$, and B-spline approximation $\hat{s}$ used in the proof of Lemma A.3.

For the two surfaces $s_1$ and $s_2$, we first find two continuous functions $\bar{s}_1$ and $\bar{s}_2$ for approximation. Specifically, $\bar{s}_1$ and $\bar{s}_2$ are interpolations of sampled data on $s_1$ and $s_2$ with $N_i$ grids along $i$-th dimension, $i = 1, 2, \cdots, n$. Since Assumption 4.2 holds, from Lemma A.2 we know that there exist $d \in \mathbb{N}^+$, $\ell_i \in \mathbb{N}^+$ and control points $C_1$ and $C_2$ of dimension $\ell_1 \times \ell_2 \times \cdots \times \ell_n$ such that $\|\hat{s}_1 - \bar{s}_1\|_2 < \epsilon_1/2$, $\|\hat{s}_2 - \bar{s}_2\|_2 < \epsilon_2/2$. We also know that the optimal control points are obtained by solving the following least square (LS) problem to fit the sampled data on $s_1$ and $s_2$.

$$
\begin{aligned}
\sum_{i_1=1}^{\ell_1} \sum_{i_2=1}^{\ell_2} \cdots \sum_{i_n=1}^{\ell_n} c_{1,i_1,i_2,\cdots,i_n} B_{i_1,d_1}(x_1) B_{i_2,d_2}(x_2) \cdots B_{i_n,d_n}(x_n) = s_1(x_1, x_2, \cdots, x_n), \forall x \in \mathcal{D}_1, \\
\sum_{i_1=1}^{\ell_1} \sum_{i_2=1}^{\ell_2} \cdots \sum_{i_n=1}^{\ell_n} c_{2,i_1,i_2,\cdots,i_n} B_{i_1,d_1}(x_1) B_{i_2,d_2}(x_2) \cdots B_{i_n,d_n}(x_n) = s_2(x_1, x_2, \cdots, x_n), \forall x \in \mathcal{D}_2,
\end{aligned}
\tag{59}
$$

where $\mathcal{D}_1$ and $\mathcal{D}_2$ are the sets of all sampled data on $s_1$ and $s_2$ and are assumed to be sufficiently dense on $\Omega$. Then, we can write the LS problem into the matrix form as follow.

$$
\begin{aligned}
A_1 C_1 &= b_1, \\
A_2 C_2 &= b_2,
\end{aligned}
\tag{60}
$$

where $A_1 = A_2$ and $\|b_1 - b_2\|_2 < \delta'(\epsilon)$, since $s_1$ and $s_2$ are Lipschitz on the bounded domain with $\|s_1 - s_2\|_2 < \epsilon$. Here $\delta'(\epsilon) \to 0$ as $\epsilon \to 0$. Without loss of generality, we consider cases where the sampled data on $s_1$ and $s_2$ is dense enough such that both $A_1$ and $A_2$ are full rank. Then by results from the LS problems with perturbation (Wei, 1989), we know that the difference of the LS solutions of the two problems in equation 60 is bounded by

$$
\|C_1 - C_2\| < \delta(\epsilon),
\tag{61}
$$

where $\delta(\epsilon) \to 0$ as $\epsilon \to 0$.

Since $s_1$ and $s_2$ are continuous functions defined on bounded domain, we know that both functions are Lipschitz. We denote $L_i$ the larger Lipschitz constants of the two functions along dimension $i = 1, 2, \cdots, n$, i.e., $\forall x = [x_1, x_2, \cdots, x_n], x' = [x'_1, x'_2, \cdots, x'_n] \in \mathbb{R}^n$,

$$
\begin{aligned}
|s_1(x) - s_1(x')| &\leq L_1 |x_1 - x'_1| + L_2 |x_2 - x'_2| + \cdots + L_n |x_n - x'_n|, \\
|s_2(x) - s_2(x')| &\leq L_1 |x_1 - x'_1| + L_2 |x_2 - x'_2| + \cdots + L_n |x_n - x'_n|.
\end{aligned}
\tag{62}
$$

Then we know

$$
\begin{aligned}
\|s_1 - \bar{s}_1\|_2 &\leq \frac{L_1(b_1 - a_1)}{N_1} + \frac{L_2(b_2 - a_2)}{N_2} + \cdots + \frac{L_n(b_n - a_n)}{N_n}, \\
\|s_2 - \bar{s}_2\|_2 &\leq \frac{L_1(b_1 - a_1)}{N_1} + \frac{L_2(b_2 - a_2)}{N_2} + \cdots + \frac{L_n(b_n - a_n)}{N_n},
\end{aligned}
\tag{63}
$$

since $\bar{s}_1$ and $\bar{s}_2$ are the interpolations of sampled data on $s_1$ and $s_2$ with $N_i$ grids along $i$-th dimension. We know that $\|s_1 - \bar{s}_1\|_2 \to 0$ and $\|s_2 - \bar{s}_2\|_2 \to 0$ when $N_i \to \infty$ for all $i$. Thus, we can find $N_1, N_2, \cdots, N_n$

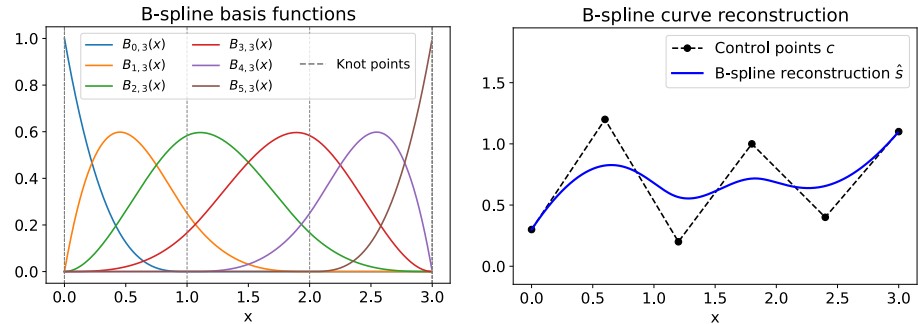

Figure 8: Visualization of B-spline basis functions (left) and B-spline reconstruction (right). Note that with the clamped knot points, $B_{0,3}(0) = B_{5,3}(3) = 1$, so that the reconstruction $\hat{s}(0) = c_1$ and $\hat{s}(3) = c_\ell$ to enforce hard boundary satisfaction.

such that $\|s_1 - \bar{s}_1\|_2 < \epsilon_1/2$, $\|s_2 - \bar{s}_2\|_2 < \epsilon_2/2$. Then by the triangle inequality we have

$$\begin{aligned}
\|s_1 - \hat{s}_1\|_2 &\leq \|s_1 - \bar{s}_1\|_2 + \|\hat{s}_1 - \bar{s}_1\|_2 < \epsilon_1/2 + \epsilon_1/2 = \epsilon_1, \\
\|s_2 - \hat{s}_2\|_2 &\leq \|s_2 - \bar{s}_2\|_2 + \|\hat{s}_2 - \bar{s}_2\|_2 < \epsilon_2/2 + \epsilon_2/2 = \epsilon_2.
\end{aligned} \tag{64}$$

$\square$

## C  Additional Results

### C.1  Convex Hull Property of B-Splines

Considering a one-dimensional B-spline of the form as $\hat{s}(x) = cB_d(x)$, where $x \in [a, b]$, we have

$$\hat{s} \in [a, b] \to [\underline{c}, \overline{c}], \tag{65}$$

where

$$\underline{c} = \min_{i=1,\dots,\ell} c_i, \qquad \overline{c} = \max_{i=1,\dots,\ell} c_i.$$

This property is inherent to the Bernstein polynomials used to generate Bézier curves. Specifically, the Bézier curve is a subtype of the B-spline, and it is also possible to transform Bézier curves into B-splines and vice versa (Prautzsch, 2002).

This property also holds in the multidimensional case when the B-spline is represented by a tensor product of the B-spline basis functions in equation 7 (Prautzsch, 2002):

$$\hat{s} \in [a_1, b_1] \times \cdots \times [a_n, b_n] \to [\underline{c}, \overline{c}], \tag{66}$$

where

$$\underline{c} = \min_{\substack{i_1=1,\dots,\ell_1 \\ i_2=1,\dots,\ell_2 \\ \vdots \\ i_n=1,\dots,\ell_n}} c_{i_1,i_2,\dots,i_n}, \qquad \overline{c} = \max_{\substack{i_1=1,\dots,\ell_1 \\ i_2=1,\dots,\ell_2 \\ \vdots \\ i_n=1,\dots,\ell_n}} c_{i_1,i_2,\dots,i_n}.$$

This property offers a practical tool for verifying the reliability of the results produced by the trained learning scheme. In the case of learning recovery probabilities, the approximated solution should provide values between 0 and 1. Since the number of control points is finite, a robust and reliable solution occurs if all generated control points are within the range $[0, 1]$, i.e.,

$$\underline{c} = 0 \qquad \overline{c} = 1.$$

### C.2 B-Spline Representation for General Domains

In this section, we present results on using B-splines to represent a surface in arbitrary domains. Without loss of generality, we consider 2D cases. Specifically, we consider a original domain $(u, v) \in D \triangleq [0, 1] \times [0, 1]$, and a surface defined on a parametric domain via the following transformation

$$
\begin{aligned}
x(u, v) &= g_x(u, v), \\
y(u, v) &= g_y(u, v), \\
z(u, v) &= f(x(u, v), y(u, v)).
\end{aligned}
$$

Note that since $g_x$ and $g_y$ are arbitrary, $(x, y)$ can lie in arbitrary domains of interest.

We approximate the surface by reconstructing $x$, $y$ and $z$ using tensor-products B-spline defined on $(u, v)$, and with slight abuse of notation, we use $s$ to denote the approximation:

$$
s(u, v) = \sum_{i=1}^{n_u} \sum_{j=1}^{n_v} \mathbf{c}_{ij} B_{i,d}(u) B_{j,d}(v), \quad s(u, v) \in \mathbb{R}^3,
$$

where the control points $\mathbf{c}_{ij}$ are given by

$$
\mathbf{c}_{ij} = \begin{bmatrix} c_{ij}^x \\ c_{ij}^y \\ c_{ij}^z \end{bmatrix} \in \mathbb{R}^3,
$$

and $B_{i,d}(u)$ and $B_{j,d}(v)$ are B-spline basis functions of degrees $d$ in the $u$ and $v$ directions, respectively, defined on clamped knot vectors as in Section 3.2. The vectors $\mathbf{c}_{ij}$ are the control points in $\mathbb{R}^3$. With this representation, the surface components can be written as the following three separate B-spline functions

$$
\begin{aligned}
s_x(u, v) &= \sum_{i=1}^{n_u} \sum_{j=1}^{n_v} c_{ij}^x B_{i,d}(u) B_{j,d}(v), \\
s_y(u, v) &= \sum_{i=1}^{n_u} \sum_{j=1}^{n_v} c_{ij}^y B_{i,d}(u) B_{j,d}(v), \\
s_z(u, v) &= \sum_{i=1}^{n_u} \sum_{j=1}^{n_v} c_{ij}^z B_{i,d}(u) B_{j,d}(v).
\end{aligned}
$$

While the $(u, v)$ parameters lie on a regular meshgrid, the actual surface domain in the $(x, y)$ space is shaped through the mapping functions $g_x$ and $g_y$, and can also be approximated by B-splines.

To sample points on the surface, we can evaluate a meshgrid on $D$ and compute:

$$
\begin{aligned}
x_{k\ell} &= g_x(u_k, v_\ell), \\
y_{k\ell} &= g_y(u_k, v_\ell), \\
z_{k\ell} &= f(x_{k\ell}, y_{k\ell}),
\end{aligned}
$$

where $k = 1, \ldots, N_u$ and $\ell = 1, \ldots, N_v$ are the indices of sampled points in the $u$ and $v$ directions, respectively.

Similarly, the control points are arranged on a meshgrid over $D$. Due to the clamped nature of the B-spline basis functions, the first and last basis functions in each direction satisfy:

$$
\begin{aligned}
B_{1,d}(0) &= 1, \quad B_{n_u,d}(1) = 1, \\
B_{1,d}(0) &= 1, \quad B_{n_v,d}(1) = 1,
\end{aligned}
$$

and all other basis functions vanish at these boundary values. This property allows direct control over the surface boundary through the corresponding boundary control points. For example, we can write:

$$s_x(v)\big|_{u=0} = \sum_{j=1}^{n_v} c_{1j}^x B_{j,d}(v), \tag{67}$$

$$s_x(v)\big|_{u=1} = \sum_{j=1}^{n_v} c_{n_u j}^x B_{j,d}(v), \tag{68}$$

$$s_x(u)\big|_{v=0} = \sum_{i=1}^{n_u} c_{i1}^x B_{i,d}(u), \tag{69}$$

$$s_x(u)\big|_{v=1} = \sum_{i=1}^{n_u} c_{in_v}^x B_{i,d}(u). \tag{70}$$

The same one-dimensional B-spline representation applies to the $y$ and $z$ components. This demonstrates that a subset of control points governs the shape and values of the surface along the boundary.

This method generalizes naturally to higher-dimensional problems, since each surface component is represented independently using B-spline functions.

In the following, we show an example of approximating a surface on an annulus with the methods above. We consider a parametric domain $(u, v) \in [0, 1] \times [0, 1]$, and define a transformation to an annular domain in $\mathbb{R}^2$ with inner radius $r_{\mathrm{inner}} = 1$ and outer radius $r_{\mathrm{outer}} = 2$. The radial coordinate $R(u)$ is defined as a convex combination of squared radii

$$R(u) = \sqrt{(1 - u)r_{\mathrm{inner}}^2 + u r_{\mathrm{outer}}^2}.$$

The angular coordinate is set as $\theta(v) = 2\pi v$. The transformation to Cartesian coordinates is then given by

$$x(u, v) = R(u)\cos(2\pi v),$$
$$y(u, v) = R(u)\sin(2\pi v).$$

This maps the unit square $[0, 1]^2$ to an annular region in $(x, y)$-space.

We define a scalar height function $z(x, y)$ over the annulus, depending on the transformed radial and angular coordinates

$$z(u, v) = 0.3\sin(3R(u))\cos(10\pi v).$$

Hence, we can represent the surface $z$ on $(x, y)$ with the $(u, v)$ coordinates as follows.

$$\begin{bmatrix} x(u, v) \\ y(u, v) \\ z(u, v) \end{bmatrix} = \begin{bmatrix} R(u)\cos(2\pi v) \\ R(u)\sin(2\pi v) \\ 0.3\sin(3R(u))\cos(10\pi v) \end{bmatrix}.$$

We then generate data on a $100 \times 100$ equispaced grid in the $(u, v)$ domain, and fit a B-spline surface of order 3 using 30 control points along each direction. We use this grid structure to independently learn a B-spline surface for each of the $x$, $y$, and $z$ coordinates, allowing us to represent the geometry of the annular surface. The boundary conditions are enforced by separately computing the approximating B-splines in equation 67 for each component along the boundary. To determine the remaining control points, we solve a least squares problem based on data points.

Fig. 9 visualizes the results. We can see that the B-splines are able to accurately reconstruct the surface on the annulus. Note that the transformation from $(u, v)$ to $(x, y)$ could be arbitrary other than the annulus, suggesting the possibility of future work to learn PDE solutions in arbitrary domains.

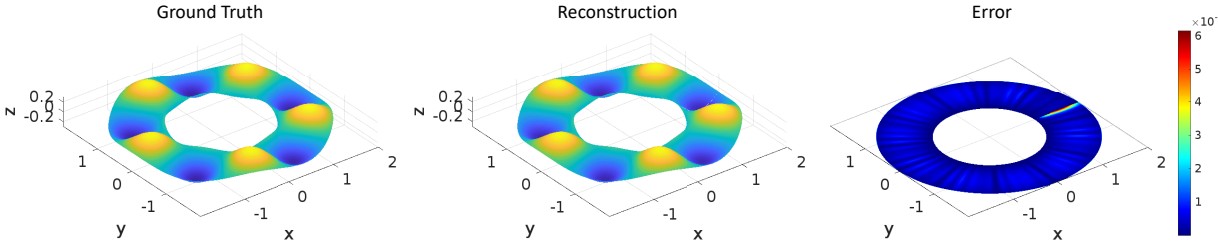

Figure 9: B-spline reconstruction of a surface on an annulus.

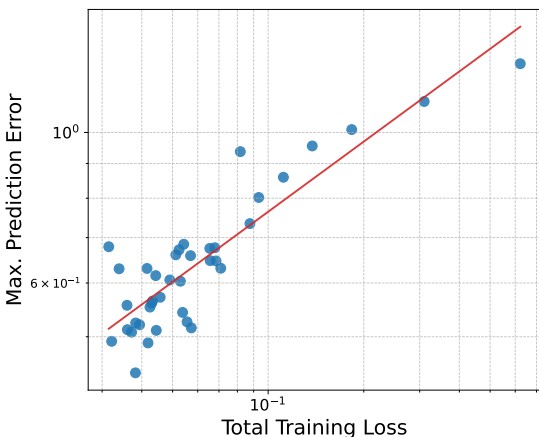

Figure 10: Maximum prediction error vs. total training loss for advection equations.

## C.3 Generalization Error Verification

In this section, we conduct experiments to verify the derived generalization error bounds in Theorem 4.6. Specifically, we consider the settings for the advection equation experiments in Section 5.2. We train PI-BSNet for 2000 epochs, sample results at every 50 epochs, and visualize the *maximum* prediction error over the PDE family (30 parametric PDE instances) vs. total training loss in Fig. 10. It can be seen that the maximum prediction error is generally linearly depedent on the total training loss, which matches the provided theoretical results.

## C.4 Other Boundary Condition Types

In this section, we provide experiment results for parametric PDEs with Neumann type boundary conditions. Specifically, we consider the following parametric diffusion equation

$$\frac{\partial s}{\partial t}(x,t) - u\frac{\partial^2 s}{\partial x^2}(x,t) = 0, \tag{71}$$

with IC

$$s(x,0) = \cos(\pi x), \tag{72}$$

and the following Neumann BCs on both sides

$$\frac{\partial s}{\partial x}(0,t) = 0, \quad \frac{\partial s}{\partial x}(1,t) = 0. \tag{73}$$

The equation has the analytical solution as follows

$$s(x,t) = \cos(\pi x)e^{-u\pi^2 t}. \tag{74}$$

Table 1: Training time and average test relative $L_2$ error for the Neumann parametric diffusion problem.

| Method | Training time (s) | Avg. test relative $L_2$ error |
|---|---|---|
| PINN | 914.4 | $1.697 \times 10^{-2}$ |
| PI-DeepONet | 1064.5 | $4.459 \times 10^{-2}$ |
| PI-BSNet | 148.6 | $1.932 \times 10^{-2}$ |

---

**Algorithm 1** Training Physics-Informed Deep B-Spline Networks (PI-BSNet)

---

**Given:** Training data $\mathcal{D} = \{(u, \alpha, s)\}_{N_{\text{train}}}$
**Input:** B-spline order $d$; control points number $\ell$; loss weights $w_p$, $w_d$, $w_b$; epochs $E$
**Output:** Trained PI-BSNet parameters $\boldsymbol{\theta}^*$

1: Compute B-spline basis $B_d$ and derivatives $B_d'$, $B_d''$ and higher derivatives if needed
2: **for** epoch $= 1$ to $E$ **do**
3:    **for** each sample $(u, \alpha, s) \in \mathcal{D}$ **do**
4:       **Predict control points:** $\tilde{C} \leftarrow G_{\boldsymbol{\theta}}(u, \alpha)$
5:       **Assign ICBCs:**
        $\tilde{C}[0, :] \leftarrow$ IC, $\tilde{C}[:, \text{end}] \leftarrow$ BC   ▷ Specific control points assignment depending on actual ICBCs
6:       **Assemble B-spline approximation:** $\tilde{s}(x) = G_{\boldsymbol{\theta}}(u, \alpha)(x)$ via equation 7
7:       **Evaluate physics residual:**
8:         $\partial_x \tilde{s}(x)$, $\partial_{xx} \tilde{s}(x)$ using $B_d$, $B_d'$, $B_d''$ via equation 12
9:         $\mathcal{F}(\tilde{s}, x, u)$ for all $x \in \mathcal{P}$
10:      **Compute losses:**
11:        $\mathcal{L}_p = \frac{1}{|\mathcal{P}|} \sum_{x \in \mathcal{P}} |\mathcal{F}(\tilde{s}, x, u)|^2$
12:        $\mathcal{L}_d = \frac{1}{|\mathcal{D}|} \sum_{x \in \mathcal{D}} |s(x) - \tilde{s}(x)|^2$
13:        If needed: $\mathcal{L}_b = \frac{1}{|\mathcal{M}|} \sum_{x \in \mathcal{M}} |\mathcal{B}(\tilde{s}, x, u)|^2$
14:      **Update parameters:**
15:        optimize $\boldsymbol{\theta}$ with loss $\mathcal{L} = w_p \mathcal{L}_p + w_d \mathcal{L}_d + w_b \mathcal{L}_b$
16: **return** $\boldsymbol{\theta}^*$

---

We consider $u \in [0.1, 1.5]$ and generated 50 training data and 10 testing data with uniformly randomly sampled $u$. We train the proposed PI-BSNet with number of control points $\ell_x = \ell_t =$, with B-spline order $d = 5$ for 5000 epochs. The data loss, PDE loss, and ICBC loss weights are set to 5, 1, 2 respectively. Note that we impose the Neumann BC as a loss function too since B-spline does not support direct assignment for derivatives. The coefficient network is a 3 layer fully connected NN with 128 hidden width at each layer. We compare the proposed PI-BSNet with PINN and PI-DeepONet with similar architectures under the same setting. Fig. 11 shows the prediction visualization for different PDE parameter $u$, for all three methods. The training time and average prediction $L_2$ errors are reported in Table 1. It can be seen that the proposed PI-BSNet achieves similar performance as baselines, but is much faster due to the compact representation.

## D   Experiment Details

In this section, we provide details for the experiments in Section 5. The general procedures for training the proposed PI-BSNet is shown in Alg. 1. In all statistical results, test trials were selected by uniformly random sampling from the corresponding parameter spaces. All methods were evaluated on the same test sets to ensure a fair comparison.

### D.1   Recovery Probabilities

**Training Data:** The convection diffusion PDE defined in equation 17 and equation 18 has analytical solution

$$s(x, t) = \int_0^t \frac{(\alpha - x)}{\sqrt{2\pi\tau^3}} \exp\left(-\frac{((\alpha - x) - u\tau)^2}{2\tau}\right) d\tau, \tag{75}$$

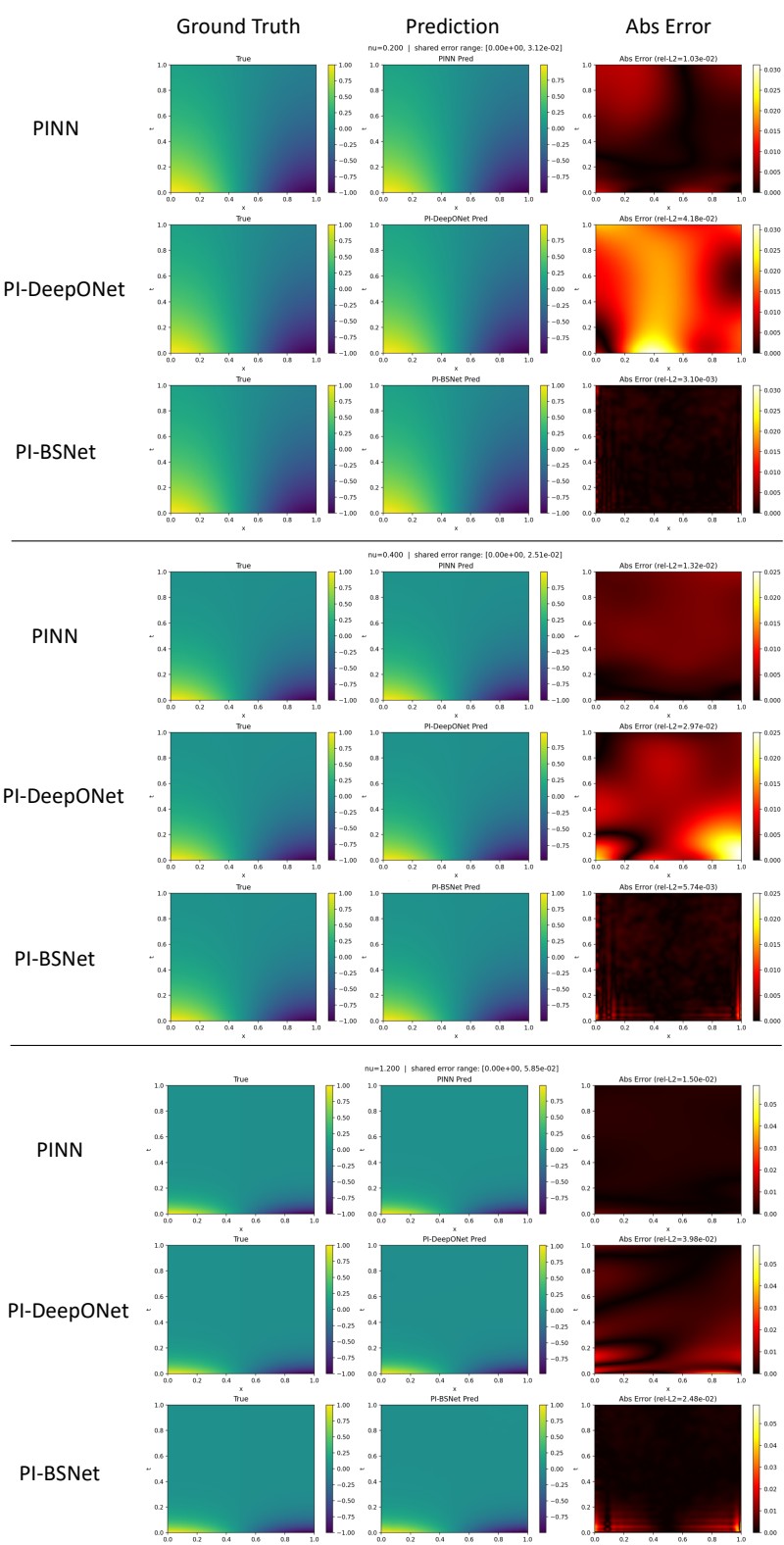

Figure 11: Test results on diffusion equations with Neumann boundary conditions.

where $\alpha$ is the parameter of the boundary of the set $\mathcal{C}_\alpha = \{x \in \mathbb{R} : x \geq \alpha\}$, and $u$ is the parameter in the system dynamics $dx_t = u\,dt + dw_t$. We use numerical integration to solve equation 75 to obtain ground truth training data for the experiments. We uniformly sample 40 random $u \in [0, 2]$ and $\alpha \in [0, 4]$ for training data generation.

**Network Configurations:**

- **PI-BSNet (proposed):** We use 3-layer fully connected neural networks with ReLU activation functions for the coefficient network. The number of neurons for each hidden layer is set to be 64. We use Adam as the optimizer.

- **PINN** (Raissi et al., 2019): A 3-layer fully connected neural network with Tanh activations is used to approximate the solution directly. The PDE residuals are computed via automatic differentiation. Initial and boundary conditions losses are imposed.

- **EPINN** (Wang et al., 2023): A 3-layer fully connected network with Tanh activation is used. The exact solution form is preserved via the function template $g + \phi_x \phi_t f_\theta$, where $f_\theta$ is approximated by the network. The spatial-temporal encoding is based on domain-aware transformations using $\phi_x = \frac{a-x}{a-x_{\min}}$ and $\phi_t = \frac{t}{T}$.

- **SFHCPINN** (Li et al., 2024a): Extends EPINN by applying sinusoidal feature embeddings (SIREN-style) to the input coordinates $(x, t)$ before passing them into the network. These features are linearly transformed and concatenated with parameters $(\lambda, a)$ before being passed to a 2-layer MLP with ReLU activations. The template $g + \phi_x \phi_t f_\theta$ is still enforced.

- **HWPINN** (Chen et al., 2023): The same formulation $g + \phi_x \phi_t f_\theta$ is used as in EPINN while the standard hidden layer width is replaced by a wider fully connected MLP (two hidden layers of 64 neurons each with Tanh activation). This model emphasizes expressive capacity within the hard-constraint formulation.

- **VS-PINN** (Ko & Park, 2025): A learnable input transformation is applied to $(x, t)$ via trainable scaling factors before the hard-constrained template is enforced. This adds a degree of flexibility to adapt to unknown domain warping or scale mismatch, while preserving the $g + \phi_x \phi_t f_\theta$ form.

- **CAN-PINN** (Chiu et al., 2022): The hard-constrained formulation is used along with a finite-difference approximation to compute second-order spatial derivatives in the PDE residual. The model uses a standard MLP with two hidden Tanh layers (64 neurons each) and predicts the residual via custom finite difference steps.

- **PI-DeepONet** (Goswami et al., 2023): A DeepONet structure is used with separate branch and trunk networks, each comprising two Tanh layers of 64 neurons. The outputs of the two networks are fused through an inner product after a linear transformation, enforcing parameter conditioning and spatial-temporal separation. Initial and boundary condition losses are imposed.

- **PI-FNO** (Li et al., 2024b): This model is implemented based on the Tensorized Fourier Neural Operator (TFNO2d) module, and it operates on a 4-channel input grid of shape $[\lambda, a, x, t]$. The architecture includes 4 spectral convolution layers with 12 Fourier modes in each direction and a hidden width of 32. PDE residuals are computed using automatic differentiation on the input grid, and ICBC losses are imposed.

Note that we extend the original PINN framework (Raissi et al., 2019) by augmenting the network inputs with the PDE and ICBC parameters $u$ and $\alpha$, enabling the model to learn mappings to solutions over families of parametrized PDEs. Similar extensions are applied to other PINN variants used as baselines to ensure a fair comparison.

**Full Visualization:** Visualization of prediction results for all methods is shown in Fig. 12.

### D.2 Advection Equations

**Training Data:** The advection equation defined in equation 19 admits an analytical solution

$$s(x, t) = \sin\left(2\pi\left((x - ut) \bmod 1\right) + \alpha\right),$$

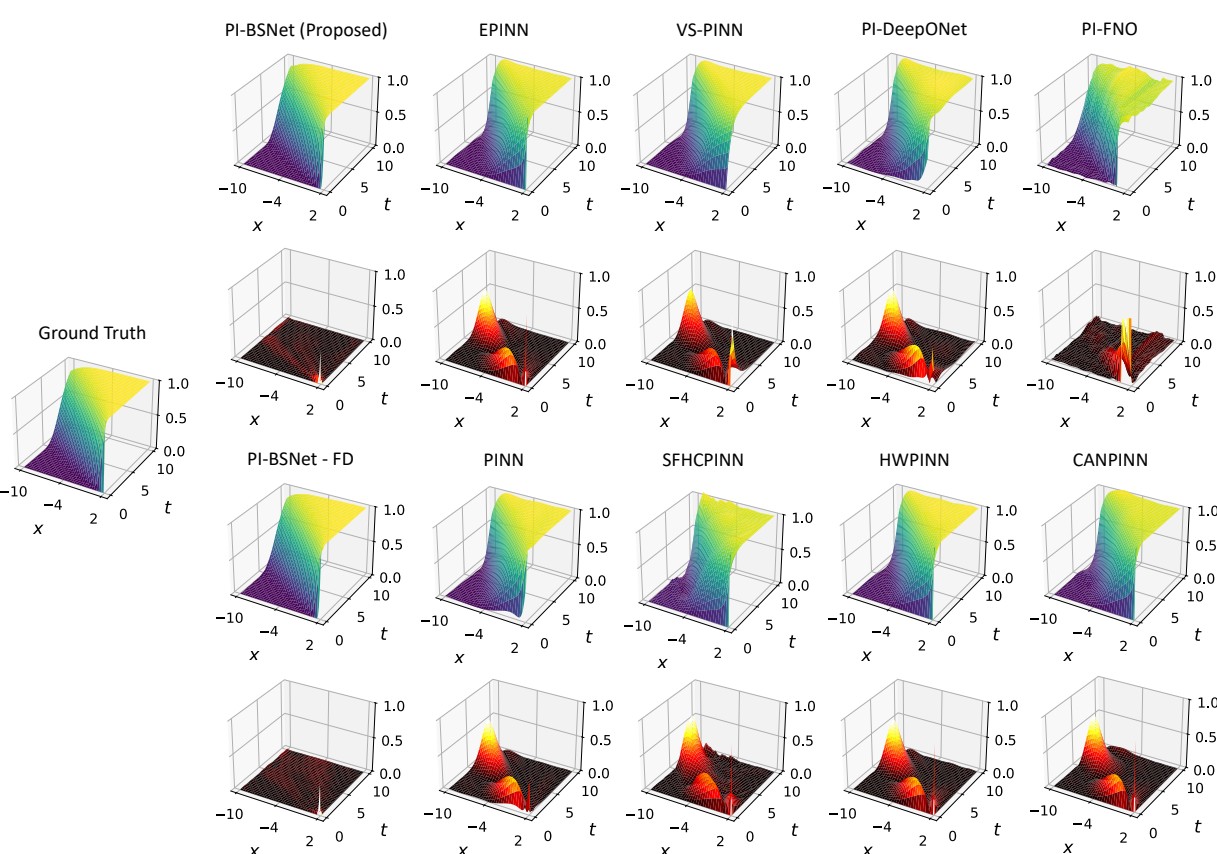

Figure 12: Recovery probability visualizations. Predictions in first row and errors in second row.

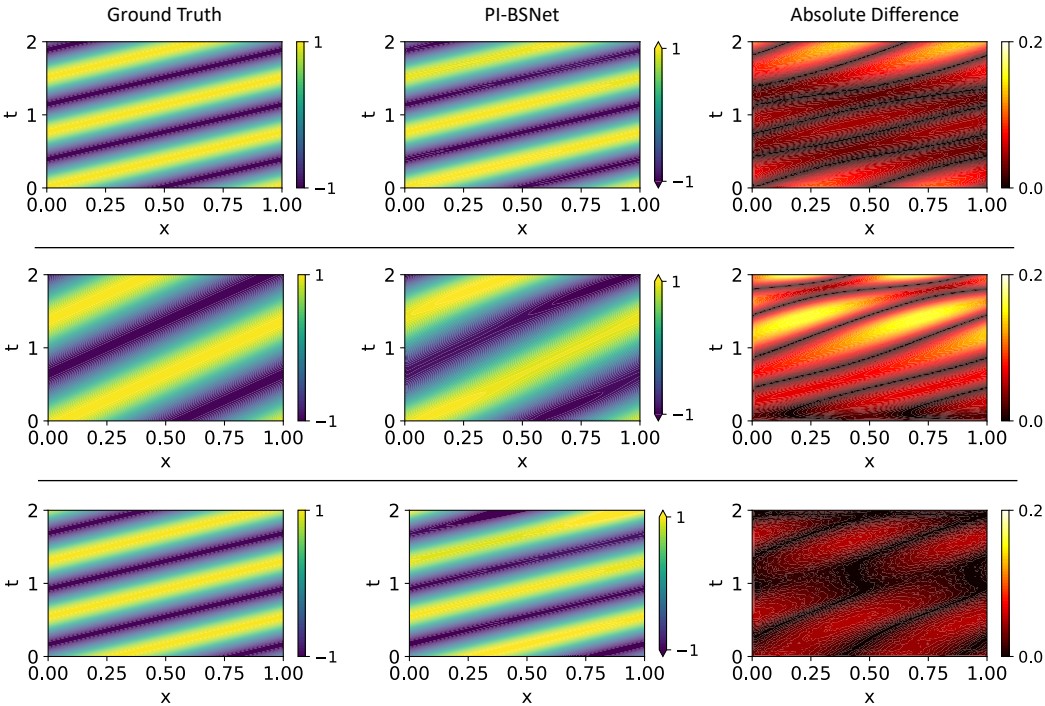

Figure 13: Results on advection equation with different random parameters.

which is evaluated on a uniform spatio-temporal grid with 100 points in $x \in [0,1]$ and 100 points in $t \in [0,2]$. For each of the 100 training samples, we uniformly sample $u \in [0.5, 1.5]$ and $\alpha \in [0, 2\pi)$.

**Network Configurations:** We train PI-BSNet with 3-layer fully connected neural networks with ReLU activation on varying parameters $u \in [0.5, 1.5]$ and $\alpha \in [0, 2\pi)$, and test on randomly selected parameters in the same domain. The B-spline basis of order 5 is used and the number of control points along $x$ and $t$ are set to be $\ell_x = \ell_t = 150$. Note that more control points are used in this case study to represent the high frequency solution.

**Additional Visualizations:** Additional visualizations for testing results with different random parameters are shown in Fig. 13.

### D.3 Diffusion on Trapezoid

**Domain Transformation:** The exit probability $s(x, y, t)$ defined in equation 22 is the solution of the following diffusion equation

$$\frac{\partial s}{\partial t} = \frac{1}{2}\left(\frac{\partial^2 s}{\partial x^2} + \alpha \frac{\partial^2 s}{\partial y^2}\right), \tag{76}$$

where $\alpha \in [0, 2]$ is an unknown parameter. And the ICBCs are

$$\begin{aligned}
s(0, x, y) &= 0, \quad \forall (x, y) \in \Omega_{\text{target}}, \\
s(t, x, y) &= 1, \quad \forall t \in [0, T], \ \forall (x, y) \in \partial\Omega_{\text{target}},
\end{aligned} \tag{77}$$

We define the square mapped domain as

$$\Omega_{\text{mapped}} = \{(u, v) \in \mathbb{R}^2 : u \in [0, 1], \ v \in [0, 1]\}, \tag{78}$$

and we can find the mapping from the target domain to this mapped domain as

$$(u, v) = T(x, y) = \left(\frac{x + 1 - 0.5y}{2 - y}, \ y\right), \tag{79}$$

which maps the left boundary $x = -1 + 0.5y$ of $\Omega_{\text{target}}$ to the left edge $u = 0$ of $\Omega_{\text{mapped}}$ and the right boundary $x = 1 - 0.5y$ to the right edge $u = 1$, while preserving $y$. The inverse mapping is then given by

$$(x, y) = T^{-1}(u, v) = (-1 + 0.5v + (2 - v)u, \ v). \tag{80}$$

Note that the mapped domain $\Omega_{\text{mapped}}$ can be readily handled by PI-BSNet. We then derive the transformed PDE on the mapped square domain $\Omega_{\text{mapped}}$ to be

$$\frac{\partial s}{\partial t} = \frac{1}{2} \frac{1}{2 - v} \left[ \frac{\partial}{\partial u} \left( A(u,v) \frac{\partial s}{\partial u} - B(u,v) \frac{\partial s}{\partial v} \right) + \frac{\partial}{\partial v} \left( -B(u,v) \frac{\partial s}{\partial u} + D(u,v) \frac{\partial s}{\partial v} \right) \right], \tag{81}$$

where

$$A(u,v) = \frac{1 + \alpha(u - 0.5)^2}{2 - v}, \quad B(u,v) = \alpha(0.5 - u), \quad D(u,v) = \alpha(2 - v). \tag{82}$$

The corresponding ICBCs are

$$\begin{aligned} s(u, v, t) &= 1, \quad \forall t \in [0, T], \ \forall (u, v) \in \partial \Omega_{\text{mapped}}, \\ s(u, v, 0) &= 0, \quad \forall (u, v) \in \Omega_{\text{mapped}}. \end{aligned} \tag{83}$$

For efficient evaluation of the physics loss, we approximate equation 81 with the following anisotropic but cross-term-free PDE

$$\frac{\partial s}{\partial t} = \frac{1}{2} \left[ \frac{1}{(2 - v)^2} \frac{\partial^2 s}{\partial u^2} + \alpha \frac{\partial^2 s}{\partial v^2} \right]. \tag{84}$$

**Training Data:** The ground truth data is generated by solving equation 76 on the trapezoid domain using the forward Euler method, with time step size $\Delta t = 0.001$ and spatial resolution $(N_x, N_y) = (21, 21)$.

**Network Configurations:** We generate 50 sample solutions of equation 76 with varying $\alpha$ uniformly sampled from $[0, 1.5]$, and transform the solution from $\Omega_{\text{target}}$ to $\Omega_{\text{mapped}}$ via equation 79 as training data. We construct a PI-BSNet with 3-layer neural network with ReLU activation functions and 64 hidden neurons each layer. The number of control points are $\ell_x = \ell_y = 20$ and $\ell_t = 100$. The order of B-spline is set to be 3. We train the coefficient network with Adam optimizer with $10^{-3}$ initial learning rate for 3000 epoch. Note that the physics loss enforces equation 84 on $\Omega_{\text{mapped}}$, which is the domain for PI-BSNet training. We use $w_d = 1$ and $w_p = 0.001$ as the loss weights for training. We use a smaller weight for physics loss since the physics model is approximate. We then test the prediction results on unseen $\alpha$ randomly sampled from $[0, 1.5]$.

**Additional Visualizations:** Additional visualizations for testing results with different random parameters are shown in Fig. 14.

## E  Ablation Experiments

### E.1  B-spline Derivatives

In this section, we show that the analytical formula in equation 12 can produce fast and accurate calculation of B-spline derivatives. Fig. 15 shows the derivatives from B-spline analytical formula and finite difference for the 2D space $[-10, 2] \times [0, 10]$ with the number of control point $\ell_1 = \ell_2 = 15$. The control points are generated randomly on the 2D space, and the derivatives are evaluated at mesh grids with $N_1 = N_2 = 100$. We can see that the derivatives generated from B-spline formulas match well with the ones from finite difference, except for the boundary where finite difference is not accurate due to the lack of neighboring data points. Note that since there is no fitting involved, the B-spline derivatives are the ground truth values. This illustrates the advantages of using B-spline analytical derivatives over finite difference during physics-informed training.

We further consider a case where the surface is approximated by B-splines. Specifically, we consider the following 2D surface defined on $[-1, 1] \times [0, 1]$,

$$s(t, x) = \sin(2\pi x) \cos(\pi t) + 0.1 \, x^2 t + 0.05 \exp\left(-\frac{(x - 0.3)^2 + (t - 0.6)^2}{0.05}\right), \tag{85}$$

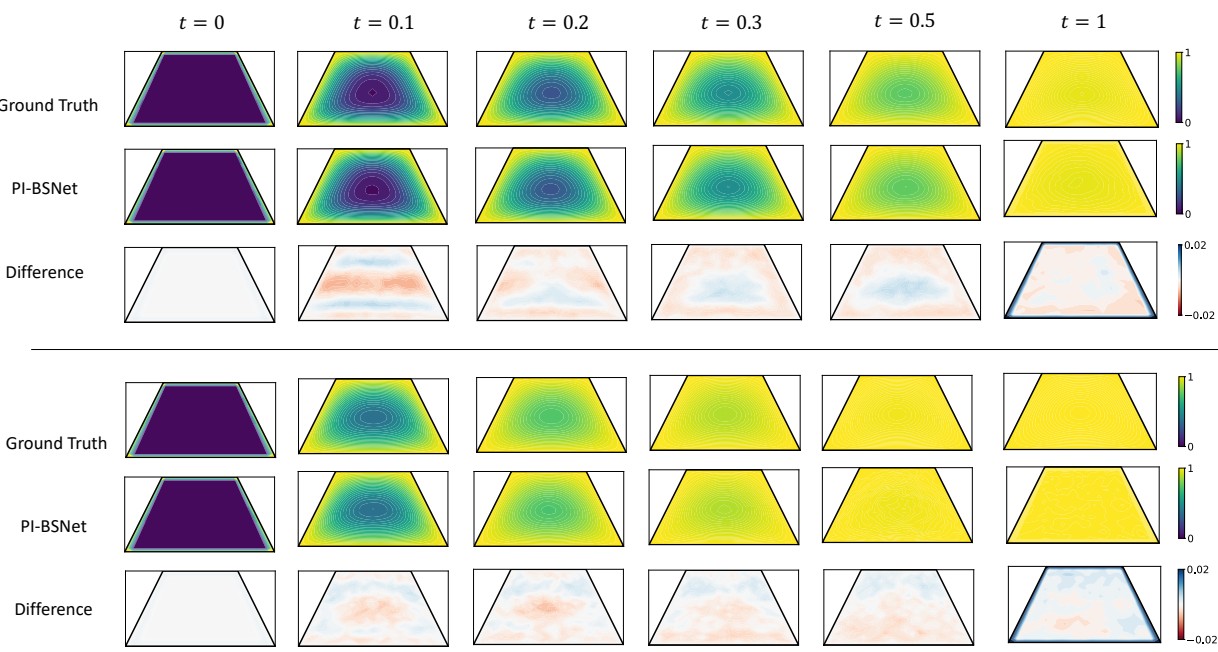

Figure 14: Results on diffusion equation on the trapezoid with different random parameters.

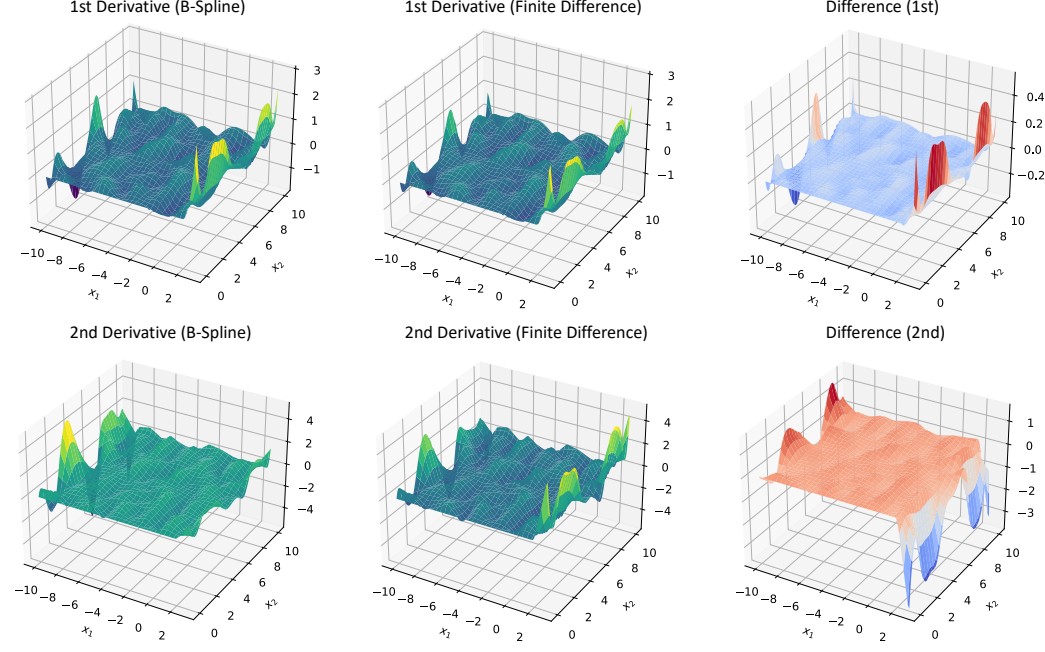

Figure 15: First and second derivatives from B-splines and finite difference.

where the ground truth gradients and Hessians can be obtained. We fit a B-spline surface with the number of control point $\ell_1 = \ell_2 = 20$ and order $d = 4$. Fig. 16 shows the derivatives from B-spline analytical formula and finite difference and Table 2 shows the computation time and errors. The derivatives are evaluated at mesh grids with $N_1 = N_2 = 100$. We can see that the derivatives generated from B-spline formulas are in general more accurate than finite difference, in this case even if the B-spline surface is approximating the ground truth surface, and the finite difference uses the ground truth data value.

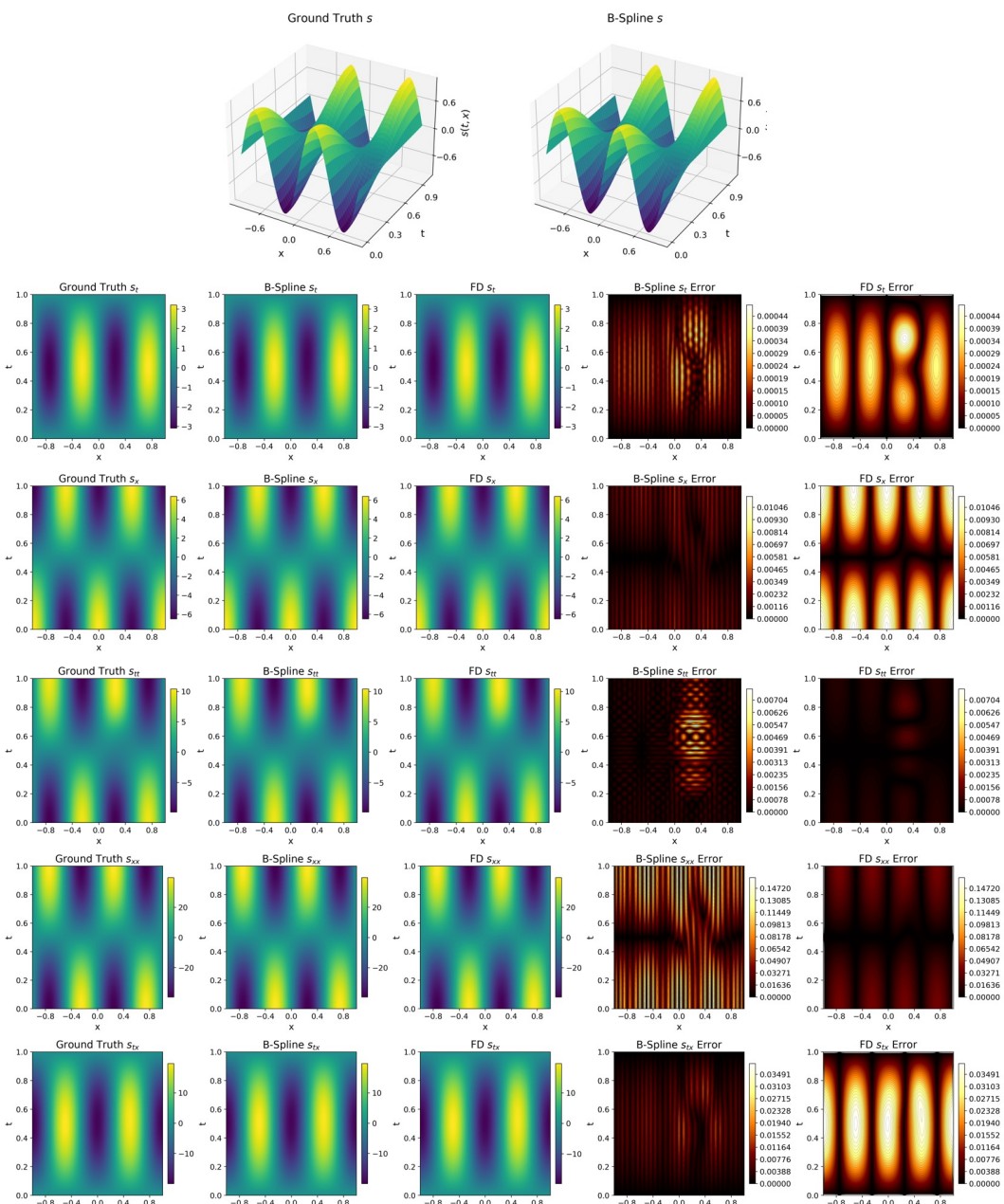

Figure 16: Derivatives from B-splines after fitting and finite difference.

Table 2: Computation time and relative $L^2$ error of the derivatives.

| Method | Time (s) | $s_t$ | $s_x$ | $s_{tt}$ | $s_{xx}$ | $s_{tx}$ |
|---|---|---|---|---|---|---|
| B-Spline | $\mathbf{4.43 \times 10^{-4}}$ | $\mathbf{6.78 \times 10^{-5}}$ | $\mathbf{4.42 \times 10^{-4}}$ | $2.23 \times 10^{-4}$ | $3.14 \times 10^{-3}$ | $\mathbf{4.59 \times 10^{-4}}$ |
| FD | $7.22 \times 10^{-4}$ | $1.17 \times 10^{-4}$ | $1.83 \times 10^{-3}$ | $\mathbf{7.53 \times 10^{-5}}$ | $\mathbf{9.15 \times 10^{-4}}$ | $1.94 \times 10^{-3}$ |

### E.2 Optimality of Control Points

In this section, we show that the learned control points of PI-BSNet are near-optimal in the $L_2$ norm sense. For the recovery probability problem considered in section 5.1, we investigate the case for a fixed set of

Table 3: Effect of ICBC enforcement strategies for PI-BSNet.

| Method | Relative $L_2$ error (mean $\pm$ std) |
|---|---|
| PI-BSNet (direct ICBC assignment) | $1.45 \pm 0.21 \times 10^{-2}$ |
| PI-BSNet (ICBC loss) | $5.52 \pm 0.55 \times 10^{-2}$ |

system and ICBC parameters $u = 1.5$ and $\alpha = 2$. We use the number of control points $\ell_1 = \ell_2 = 25$ on the domain $[-10, 2] \times [0, 10]$, and obtain the optimal control points $C^*$ in the $L_2$ norm sense by solving least square problem (Deng & Lin, 2014) with the ground truth data. We then compare the learned control points $C$ with $C^*$ and the results are visualized in Fig. 17. We can see that the learned control points are very close to the optimal control points, which validates the efficacy of PI-BSNet. The only region where the difference is relatively large is near $c_{25,0}$, where the solution is not continuous and hard to characterize with this number of control points.

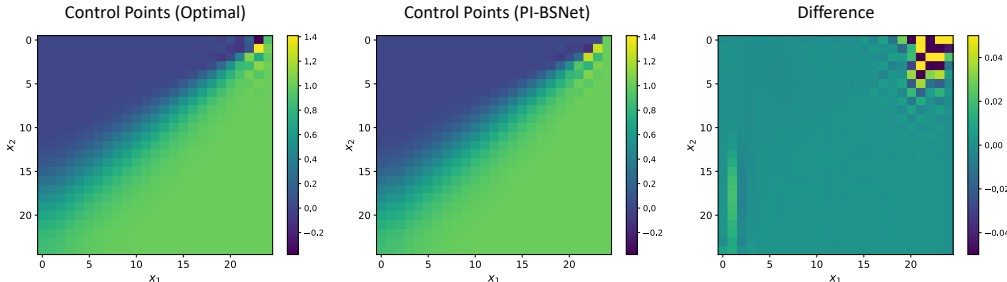

Figure 17: Control points.

### E.3   ICBC Assignment

In this section, we investigate the effect of direct ICBC assignment versus ICBC loss enforcement for the proposed PI-BSNet, and verify that explicitly assigning ICBC values through the B-spline formulation indeed benefits PDE solution learning. Specifically, we follow the experimental setting in Section 5.1 and train two variants: (i) the original PI-BSNet with direct ICBC assignment, and (ii) PI-BSNet without direct ICBC assignment, where ICBCs are imposed via an additional loss term $w_b \mathcal{L}_b$ as in equation 11. For both methods, the physics loss weight and data loss weight are set to $w_p = 1$ and $w_d = 3$, respectively, while the ICBC-loss variant uses $w_b = 3$. We repeat each setting 10 times and report the relative $L_2$ error in Table 3. The results show that introducing an additional ICBC loss degrades prediction accuracy and leads to larger errors. A representative prediction case is visualized in Fig. 18. Although enforcing ICBCs via an auxiliary loss with careful tuning can potentially yield reasonable training performance, directly assigning ICBCs is generally preferred, as balancing multiple loss terms is nontrivial and often sensitive to hyperparameter choices Wang et al. (2021a); Krishnapriyan et al. (2021).

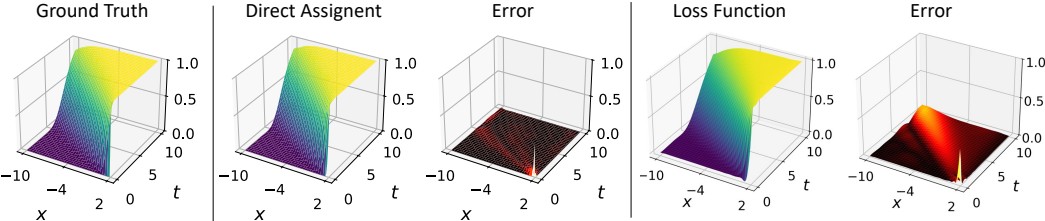

Figure 18: Results on the convection diffusion equation with variants of PI-BSNet for ICBC enforcement.

### E.4 Number of Control Points

In this section, we investigate the effect of the number of control points on the performance of PI-BSNet. The setting is described in section 5.1. Table 4 shows the approximation error and training time of PI-BSNet with different numbers of control points along each dimension. We can see that the training time increases as the number of control points increases, and the approximation error decreases, which matches with Theorem 4.3 which indicates more control points can result in less approximation error. The test-time forward pass of the model remains computationally inexpensive and exhibits negligible dependence on the number of control points.

Table 4: PI-BSNet prediction MSE with different numbers of control points along each dimension.

| Number of Control Points | 2 | 5 | 10 | 15 | 20 | 25 |
|---|---|---|---|---|---|---|
| Number of NN Parameters | 4417 | 5392 | 9617 | 17092 | 27817 | 41792 |
| Training Time (s) | 241.76 | 223.53 | 247.39 | 295.67 | 310.83 | 370.48 |
| Forward Pass Time ($\times 10^{-4}$ s) | 3.85 | 2.40 | 2.06 | 2.37 | 2.28 | 3.56 |
| Prediction MSE ($\times 10^{-4}$) | 5357.9 | 7.327 | 7.313 | 5.817 | 4.490 | 3.064 |

### E.5 Robustness and Loss Function Weights Ablations

In this section, we provide ablation experiments of the proposed PI-BSNet with different loss function configurations, and examine its robustness again noise. The setting is described in section 5.1. We first train with noiseless data and vary the data loss weight $w_d$. Table 5 shows the average MSE and its standard deviation over 10 independent runs. We can see that with more weights on the data loss, the prediction MSE reduces as noiseless data help with PI-BSNet to learn the ground truth solution. We then train with injected additive zero-mean Gaussian noise with standard deviation 0.05 and vary the physics loss weight $w_p$. Table 6 shows the results. It can be seen that increasing physics loss weights help PI-BSNet to learn the correct neighboring relationships despite noisy training data, which reduces prediction MSE. In general, the weight choices should depend on the quality of the data, the training configurations (e.g., learning rates, optimizer, neural network architecture).

Table 5: PI-BSNet prediction MSE (noiseless data).

| $w_d$ | 1 | 2 | 3 | 4 | 5 |
|---|---|---|---|---|---|
| $w_p$ | 1 | 1 | 1 | 1 | 1 |
| Prediction MSE ($\times 10^{-5}$) | $36.76 \pm 12.16$ | $12.91 \pm 10.40$ | $10.21 \pm 3.99$ | $9.28 \pm 6.78$ | $3.95 \pm 1.36$ |

Table 6: PI-BSNet prediction MSE (additive Gaussian noise data).

| $w_d$ | 1 | 1 | 1 | 1 | 1 |
|---|---|---|---|---|---|
| $w_p$ | 1 | 2 | 3 | 4 | 5 |
| Prediction MSE ($\times 10^{-4}$) | $31.58 \pm 6.46$ | $33.15 \pm 7.77$ | $13.37 \pm 11.74$ | $7.95 \pm 6.24$ | $3.86 \pm 2.05$ |

### E.6 Number of NN Layers and Parameters Ablation

In this section, we show ablation results on the number of neural network (NN) layers and parameters. We follow the experiment settings in section 5.1, and train the proposed PI-BSNet with different numbers of hidden layers, each with 10 independent runs. The number of NN parameters, the prediction MSE and its standard deviation are shown in Table 7. We can see that with 3 layers the network achieves the lowest prediction errors, while the number of layers does not have huge influence on the overall performance.

## F Additional Experiments

In this section, we provide additional experiment results.

Table 7: PI-BSNet prediction MSE with different numbers of NN layers.

| Number of Hidden Layers | 2 | 3 | 4 | 5 |
|---|---|---|---|---|
| Number of NN parameters | 37632 | 41792 | 45952 | 50112 |
| Prediction MSE ($\times 10^{-4}$) | $1.12 \pm 0.43$ | $0.90 \pm 0.42$ | $3.17 \pm 2.46$ | $3.12 \pm 2.81$ |

### F.1 Burgers' Equation

We conduct additional experiments on the following Burgers' equation, by adapting the benchmark problems in PDEBench (Takamoto et al., 2022) to account for varying system and ICBC parameters.

$$\frac{\partial s}{\partial t} + us\frac{\partial s}{\partial x} - \nu\frac{\partial^2 s}{\partial x^2} = 0, \tag{86}$$

where $\nu = 0.01$ and $u \in [0.5, 1.5]$ is a changing parameter. The domain of interest is set to be $(x, t) \in [0, 10] \times [0, 8]$, and the initial condition is

$$s(x, 0) = \exp\{-(x - \alpha)^2/2\}, \tag{87}$$

where $\alpha \in [2, 4]$ is a changing parameter. We train PI-BSNet with 3-layer fully connected neural networks with ReLU activation on varying parameters $u \in [0.5, 1.5]$ and $\alpha \in [2, 4]$, and test on randomly selected parameters in the same domain. The B-spline basis of order 4 is used and the number of control points along $x$ and $t$ are set to be $\ell_x = \ell_t = 100$. Note that more control points are used in this case study compared to the convection diffusion equation in section 5.1, as the solution of the Burgers' equation has higher frequency along the ridge which requires finer control points to represent. Adaptive basis functions can be potentially used to further reduce errors under the same number of control points. Fig. 19 visualizes the prediction results on several random parameter settings. The average relative $L_2$ error across 20 test cases is $7.294 \pm 1.908 \times 10^{-2}$.

### F.2 3D Heat Equation

We consider the 3D heat equation given by

$$\frac{\partial}{\partial t}s(x, t) = D\frac{\partial^2}{\partial x^2}s(x, t), \tag{88}$$

where $D = 0.1$ is the constant diffusion coefficient. Here $x = [x_1, x_2, x_3] \in \mathbb{R}^3$ are the states, and the domains of interest are $\Omega_{x_1} = \Omega_{x_2} = \Omega_{x_3} = [0, 1]$, and $\Omega_t = [0, 1]$. All lengths are in centimeters (cm) and the time is in seconds (s). In this experiment we solve equation 88 with random linear initial conditions:

$$s(x, t = 0) = \alpha_1 \cdot x_1 + \alpha_2 \cdot x_2 + \alpha_3 \cdot x_3 + \alpha_0 \tag{89}$$

where $\alpha_1, \alpha_2, \alpha_3 \in [-0.5, 0.5]$ and $\alpha_0 \in [0, 1]$ are randomly chosen. We impose the following Dirichlet and Neumann boundary conditions:

$$s(x, t|x_3 = 0) = s(x, t|x_3 = 1) = 1, \tag{90}$$

$$\frac{\partial}{\partial x_1}s(x, t|x_1 = 0) = \frac{\partial}{\partial x_1}s(x, t|x_1 = 1) = 0, \quad \frac{\partial}{\partial x_2}s(x, t|x_2 = 0) = \frac{\partial}{\partial x_2}s(x, t|x_2 = 1) = 0 \tag{91}$$

We set the B-splines to have the same number $\ell = 15$ of equispaced control points in each direction including time. We sample the solution of the heat equation at 21 equally spaced locations in each dimension. Thus, each time step consists of $15^3 = 3375$ control points and each sample returns $15^4 = 50625$ control points total. The inputs to our neural network are the values of $\alpha$ from which it learns the control points, and subsequently the initial condition surface via direct supervised learning. This is followed by learning the control points associated with later times, $(t > 0)$ via the PI-BSNet method. Because of the natural time evolution component of this problem, we use a network with residual connections and sequentially learn each time step. The neural network has a size of about $5 \times 10^4$ learnable parameters. We also train a standard

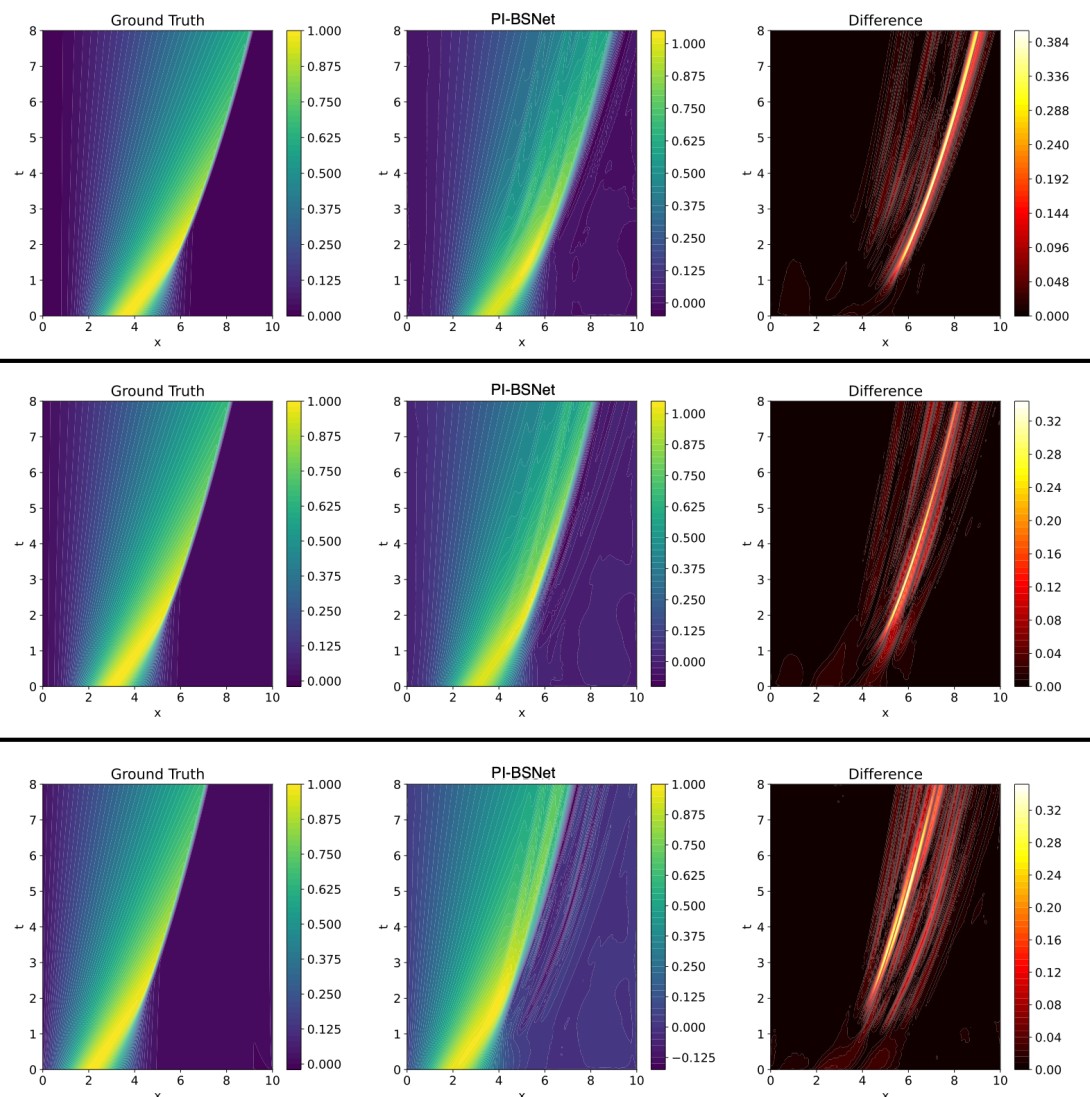

Figure 19: Results on Burgers' equations with different random parameter settings.

PINN (Raissi et al., 2019) for comparison. The PINN consists of 4 hidden layers with 50 neurons in each layer, with Tanh as the activation functions.

Fig. 20 shows the learned heat equation and a slice of the residual in the $x_1$-$t$ plane. It can be seen that with PI-BSNet the value is diffusing over time as intended. Although our initial condition does not adhere to the heat equation as estimated by the B-spline derivative, we quickly achieve a low residual, while PINN fails to do so. The average residuals during testing is 0.0121 for PINN and 0.0032 for PI-BSNet, which indicates the efficacy of the PI-BSNet method.

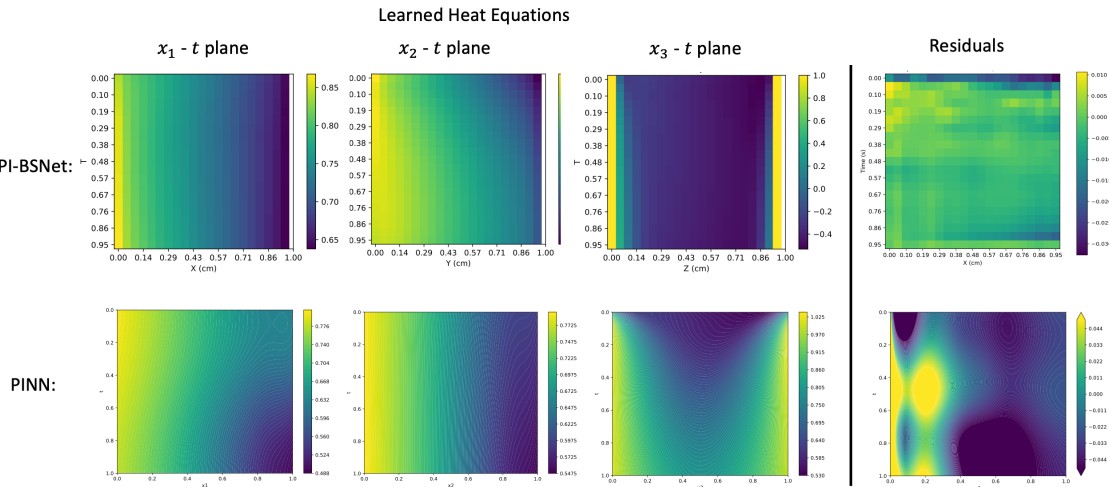

Figure 20: The learned solutions (left) and the residuals (right) for the 3D heat equations.

