# OpenReview forum: "Physics-Informed Deep B-Spline Networks"
_TMLR — Accepted by TMLR_

### Review · Reviewer_qsJy · 2025-11-19

**Summary Of Contributions:**

The paper “Physics-Informed Deep B-Spline Networks” proposes to approximate the solution to a parametric PDE with changing boundary conditions using B-Splines, which are predicted by a neural network given the values of the parameters and the specific initial condition (ICBC). The key practical contribution here lies in fixing the boundaries of the B-Splines to natively handle Dirichlet-Boundary conditions by fixing the values of the control points at the boundaries, which is especially suited for ICBCs. The dependable simplicity of this approach is a strength, but previous work already combined PINNs and Splines [1, 2, 3]. Additionally, a theoretical analysis is provided, which consists of two parts. The first part is concerned with a universal-approximation theorem, which is stated point-wise for specific instances of initial conditions and values for the parameters of the PDE and assumes a PDE with a continuous solution. It is limited in its details, as it states that for a suitable PDE with solution s, a specific instances of PDE-parameter u and boundary parameter a, there exists a neural network G that predicts control points $c = G(u,a)$ such that the corresponding spline $\tilde{s}$ is epsilon-close to $s$. But it does not state that for some epsilon and some suitable PDE there exists a G that maps u and a to c for some domain of u and a such that the corresponding $\tilde{s}$ is always epsilon-close to $s$. The second is a generalization bound between the predicted approximation $\tilde{s}$ and the true solution. This is stated in terms of the max. distance to a training-datapoint (in terms of u and a), lipschitz constants of both the NN G and the PDE (wrt to parameters u and a) and observed fit on the training examples. It demonstrates that the approach is valid, which is a strength, but it is also a very general statement. Then the experiment section follows, with 3 experiments. This section is limited as only the first experiment compares against baselines, but no other Spline-based approaches are compared to and the effect of fixing the boundary conditions is not ablated. The other two experiments have no comparisons. This is a clear weakness, a more thorough comparison would increase the quality of the contribution. These problems also appear to be non-standard.

[1] Doległo et. al, Deep neural networks for smooth approximation of physics with higher order and continuity B-spline base functions

[2] Wandel et. al, Spline-PINN: Approaching PDEs without Data using Fast, Physics-Informed Hermite-Spline CNNs

[3] Zhu et. al,  A Best-Fitting B-Spline Neural Network Approach to the Prediction of Advection–Diffusion Physical Fields with Absorption and Source Terms

**Audience:**

Yes

**Audience Explanation:**

The setting of ICBC is interesting and b-splines are tried and tested concepts with a straightforward way to fix the function values at the boundaries. The theoretical contributions on their own are probably of limited interest, as they are both very general but also not surprising given that splines with enough pieces can approximate a bounded, continuous function arbitrarily well. It is not clear whether there is something to be learned by the theoretical contributions alone.

**Broader Impact Concerns:**

no impact concerns

**Claims And Evidence:**

No

**Claims Explanation:**

The theoretical statements appear to be accurate, but in its current form the experimental section is limited in its evidence: no baselines involving splines are presented and the effect of the different choices (approximating s with splines, analytical calculation of gradients and Hessians using the splines, fixed boundaries) is not ablated. As b-splines have already been used[1], it is unclear which choice drives the differences to the other baseline. Also, most experiments do not have a thorough comparison.

[1] Doległo et. al, Deep neural networks for smooth approximation of physics with higher order and continuity B-spline base functions

**Requested Changes:**

In general, I am open to improving the assessment of the paper given improved experiments.

The critical request for changes are:
 - a more thorough comparison to existing spline-based approaches in the related-work section
 - an ablation of the different choices (b-spline, fixed boundary and analytical gradients/Hessians), showing their impact on the performance while comparing it to baselines. The experiment in E.1 for the analytical gradients/hessians is not sufficient as there are no comparisons and quantitative results, therefore the impact of the choice can not be judged
 - adding spline-based baselines (like [1, 2])

General things to improve:
 - notation in theorem 4.6: the Expectations do not mention their random variable in the function the expectation is over. This is repeated in the appendix.
 - notation in lemma A.2 (appendix): Equation 33 can be improved by not mentioning $x_1, \dots, x_n$ to make it clear that it is not point-wise at $x_1, \dots, x_n$ (so a notation like in eq 29)
 - in the proof of theorem 4.3: it would improve the writing by mentioning over what the upper bounds of control point number is
- some citations are imprecise, e.g. citing textbooks without mentioning which result is used (e.g. citation Agmon 2010 on page 21 or Pratt 2007 on page 22)
 - in equation 43: is sigma^2 here correct? it is not clear where it is coming from
 - in equation 53: I think derivation is a bit imprecise as we know that for every $x_2$ (fixed) we find $x_2’$ (knot) that is close, but then we can not choose $x_2’$ and assume it is that every $x_2$ is close. You would have to enumerate the knot-points
 - notation in eq 68: it looks like $\hat{s}$ is an element of the cartesian product between two intervals but it is a function. Similarly 69

Small things to improve:
 - in equation 63 and the following conclusion $|b_1 - b_2| < \delta’(\epsilon)$ (followed from $||s_1 - s_2||_2 < \epsilon$): It would be good to elaborate here. If I am following correctly this derives the closeness of the $s_1$ and $s_2$ evaluated on a grid from the general $l_2$ similarity of the functions, which should be due to the lipschitz properties of $s_1$ and $s_2$. As this doesn’t hold in general it would be good to write that

[1] Doległo et. al, Deep neural networks for smooth approximation of physics with higher order and continuity B-spline base functions

[2] Wandel et. al, Spline-PINN: Approaching PDEs without Data using Fast, Physics-Informed Hermite-Spline CNNs

---

> ### Author Response · Authors · 2026-01-09
> **Author Response**
>
> Thank you for your thoughtful review and constructive feedback. Please find our detailed responses to your comments below.
>
> **A more thorough comparison to existing spline-based approaches in the related-work section, and add spline-based baselines (like [1, 2]).**
>
> We thank the reviewer for suggesting spline-based baselines. We would like to clarify that the cited works [1] and [3] focus on neural network approximation of PDE solutions for fixed problem instances with data, rather than physics-informed learning of parametric solution operators. In particular, [1] and [3] employ B-spline–based neural representations to approximate smooth physical fields for a single PDE with fixed parameters and boundary conditions given data, but they do not address learning families of PDEs with varying parameters or boundary conditions, nor do they leverage physics-informed training to solve PDE instances without data. As a result, they are not applicable to the parametric PDE setting considered in this paper. Moreover, these methods do not incorporate structural enforcement of potentially varying initial and boundary conditions, nor exploit analytical derivative evaluations of the spline basis for PDE residual computation.
>
> Reference [2] (Spline-PINN) indeed combines spline representations with physics-informed learning; however, it considers a fundamentally different problem setting. Spline-PINN is designed as a forward time-marching solver for a single PDE instance with fixed parameters and boundary conditions, where the network learns a one-step temporal update and enforces PDE and boundary conditions via explicit loss terms. As a consequence, changes in physical parameters require retraining the model. In contrast, our approach learns a global space–time solution operator, for parametric PDE families with varying initial and boundary conditions (ICBCs) without re-training, and enforces certain ICBCs by construction through the spline representation. Therefore, despite superficial similarities in the use of splines and physics-informed losses, the modeling objectives, architectural design, and learning paradigms are fundamentally different.
>
> For these reasons, we hope the reviewer understands that [1–3] are not directly comparable under the considered setting. Instead, we compare our method with parametric PINN variants and physics-informed neural operator methods, which constitute more appropriate baselines for the problem addressed in this paper. We have added a detailed discussion in the Related Work section to further clarify these distinctions.
>
> [1] Doległo et. al, Deep neural networks for smooth approximation of physics with higher order and continuity B-spline base functions
>
> [2] Wandel et. al, Spline-PINN: Approaching PDEs without Data using Fast, Physics-Informed Hermite-Spline CNNs
>
> [3] Zhu et. al, A Best-Fitting B-Spline Neural Network Approach to the Prediction of Advection–Diffusion Physical Fields with Absorption and Source Terms
>
> **An ablation of the different choices (B-spline, fixed boundary, and analytical gradients/Hessians), showing their impact on the performance. The experiment on analytical gradients/Hessians is not sufficient, as there are no comparisons and quantitative results.**
>
> We thank the reviewer for this comment. We would like to clarify that the use of B-splines in PI-BSNet provides analytical expressions for spatial derivatives and Hessians (thus ground truth), which is an important advantage of the proposed formulation. In Section 5.1, we have included an ablation comparing PI-BSNet using analytical derivatives with a finite-difference variant PI-BSNet (FD), showing that both achieve similar training and prediction performance, while the finite-difference version incurs higher computational cost.
>
> In the revised Appendix E.1, we further provide a quantitative comparison of derivative errors and computation time by considering a surface that is first approximated by B-splines (results shown in Table below). We then compare derivatives computed using B-spline analytical formulas against finite-difference derivatives, where the finite-difference method is applied directly to the ground-truth surface values. The results show that B-spline–based analytical derivatives are in general more accurate and computationally more efficient, even under this favorable finite-difference setting.
>
> | Method   | Time (s)               | $s_t$               | $s_x$               | $s_{tt}$            | $s_{xx}$            | $s_{tx}$            |
> |----------|------------------------|---------------------|---------------------|---------------------|---------------------|---------------------|
> | B-Spline | $4.43 \times 10^{-4}$  | $6.78 \times 10^{-5}$ | $4.42 \times 10^{-4}$ | $2.23 \times 10^{-5}$ | $3.14 \times 10^{-3}$ | $4.59 \times 10^{-4}$ |
> | FD       | $7.22 \times 10^{-4}$  | $1.17 \times 10^{-4}$ | $1.83 \times 10^{-3}$ | $7.53 \times 10^{-5}$ | $9.15 \times 10^{-4}$ | $1.94 \times 10^{-3}$ |

---

> ### Author Response · Authors · 2026-01-09
> **Author Response (Cont.)**
>
> **In Theorem 4.6, the expectations do not explicitly indicate the random variables with respect to which they are taken. This issue also appears in the appendix.**
>
> We thank the reviewer for pointing this out. In the revised manuscript, we have clarified the notation in Theorem 4.6 and in the appendix by explicitly indicating that the expectations are taken over the random variable $x$, which is uniformly distributed on the boundary $\Omega_b(\alpha)$ and on the full domain $\Omega(\alpha)$ for the two respective conditions.
>
> **In Lemma A.2 (Appendix), the notation in Equation (33) can be improved by not mentioning $x_1,\dots,x_n$, to make it clear that the statement is not pointwise in $x_1,\dots,x_n$ (e.g., using a notation similar to Equation (29)).**
>
> We thank the reviewer for this helpful suggestion. We have revised the notation in Equation (33) of Lemma A.2 to remove the explicit dependence on $x_1,\dots,x_n$, following the notation style of Equation (29). This change clarifies that the bound is not pointwise but holds uniformly over the domain.
>
> **In the proof of Theorem 4.3, it would improve the writing by mentioning over what the upper bounds of the control point number are taken.**
>
> We thank the reviewer for this helpful suggestion. In the revised manuscript, we have clarified this point in the proof of Theorem 4.3 by explicitly stating that the control point numbers are chosen to exceed the corresponding upper bounds given in Lemma A.2, i.e., by setting $\ell_i \ge \bar{\ell}_i$ for all $i = 1,\ldots,n$. This makes clear over which bounds the control point numbers are taken and improves the clarity of the argument.
>
> **Some citations are imprecise, e.g., citing textbooks without mentioning which result is used.**
>
> We thank the reviewer for pointing this out. In the revised manuscript, we have made the citations more precise by explicitly indicating the relevant sections or chapters where the referenced results can be found. In particular, we now cite [1, Chapter 6 and Chapter 7] for the PDE regularity and stability results, and [2, Section IV-E] for the properties of B-spline representations.
>
> [1] Lawrence C. Evans. Partial differential equations, volume 19. American Mathematical Society, 2022.
>
> [2] Michael Unser, Akram Aldroubi, and Murray Eden. B-spline signal processing I: Theory. IEEE Transactions on Signal Processing, 41(2):821–833, 2002.
>
> **In equation (43), is $\sigma^2$ here correct? It is not clear where it is coming from.**
>
> Thank you for pointing this out. This was a typo in the original manuscript. The $\sigma^2$ term was not used and has been removed in the revised version.
>
> **In equation (53), the derivation is a bit imprecise. While for each fixed $x_2$ one can find a nearby knot point $x_2'$, one cannot fix a single $x_2'$ and assume it is close to all $x_2$. The knot points would need to be enumerated.**
>
> We thank the reviewer for pointing out this imprecision. We agree that a single fixed knot point cannot be used to approximate all values of $x_2$. In the revised manuscript, we explicitly enumerate the knot points and partition the domain along the $x_2$ dimension into knot-associated intervals, over which the integral is decomposed and the Lipschitz bound is applied locally before summation. We have also revised the derivation for the subsequent $x_3$ dimension in the same manner to ensure accuracy.
>
> **In Equations (68)–(69), the notation is confusing: it appears that $\hat{s}$ is treated as an element of a Cartesian product of intervals, whereas it should be a function. A similar issue appears in both equations.**
>
> We thank the reviewer for pointing this out. This was a typo and we have corrected the notation to explicitly indicate that $\hat{s}$ is a function mapping from the domain to the range, i.e., $\hat{s} : [a,b] \to [\underline{c}, \overline{c}]$ in the one-dimensional case, and $\hat{s} : [a_1,b_1] \times \cdots \times [a_n,b_n] \to [\underline{c}, \overline{c}]$ in the multidimensional case.

---

> ### Author Response · Authors · 2026-01-09
> **Author Response (Cont.)**
>
> **In equation (63) and the following conclusion $ |b_1 - b_2| < \delta'(\epsilon) $ (derived from $ \|s_1 - s_2\|_2 < \epsilon $), it would be good to elaborate. As written, this appears to infer closeness of sampled values from $L_2$ similarity of the functions, which does not hold in general without additional regularity or sampling assumptions.**
>
> We thank the reviewer for pointing out this subtle but important issue. We agree that $L_2$ closeness of two functions does not, in general, imply closeness of their sampled values. In the revised manuscript, we have clarified the assumptions under which this implication holds. Specifically, we now explicitly state that $s_1$ and $s_2$ are Lipschitz on a bounded domain and that the sampled data sets $\mathcal{D}_1$ and $\mathcal{D}_2$ are sufficiently dense on $\Omega$. Under these assumptions, discrete sample closeness follows from the continuous $L_2$ bound via standard quadrature arguments. The manuscript has been revised accordingly to make this dependence explicit.
>
>
> **Additional clarification on theoretical contributions:**
> In addition to the specific questions raised above, we would like to clarify the scope and intent of our theoretical results. The universal approximation theorem is established at the level of families of parametric PDEs, rather than for a single fixed PDE instance. Specifically, it guarantees the existence of a single coefficient network that maps PDE and ICBC parameters to B-spline control points such that the resulting spline approximation uniformly approximates the corresponding PDE solutions over a parameter domain, under mild regularity assumptions.
>
> Moreover, the generalization error theorem provides a bound on the prediction error for unseen parameter values when the model is trained on a finite set of parameter instances, without retraining. We have revised the manuscript to explicitly discuss the implications of the theoretical results, to improve clarity and readability.

---

> ### Comment · Reviewer_qsJy · 2026-01-20
> **Response to authors**
>
> I want to thank the authors for engaging with my points. I think the quality of the submission has significantly improved.
>
> **For these reasons, we hope the reviewer understands that [1–3] are not directly comparable under the considered setting. Instead, we compare our method with parametric PINN variants and physics-informed neural operator methods, which constitute more appropriate baselines for the problem addressed in this paper. We have added a detailed discussion in the Related Work section to further clarify these distinctions.**
>
> I want to thank the authors for adding the discussion in the related work section, which is crucial not to omit. I understand the author's point of not being able to directly compare to other Spline-Based approaches due to the more general parametric setting.
>
> **In the revised Appendix E.1, we further provide a quantitative comparison of derivative errors and computation time by considering a surface that is first approximated by B-splines (results shown in Table below)**
>
> I want to thank the authors for expanding the ablation experiments, it again improved the draft. The additional table provides some quantitative data, although the results appear close in accuracy and computation time. The results in 5.1 points to a benefit of the analytical approach over the finite difference in training time. But it does appear as ablating the impact of imposing dirichlet-boundary conditions remains unaddressed. An experiment pointing to a practical benefit would strengthen the case, as the fixed boundary conditions are prominently in the paper, e.g. in the abstract.
>
> **Technical Questions**
> I think all the technical questions for the proofs got sufficiently addressed.
>
> **The universal approximation theorem is established at the level of families of parametric PDEs, rather than for a single fixed PDE instance. Specifically, it guarantees the existence of a single coefficient network that maps PDE and ICBC parameters to B-spline control points such that the resulting spline approximation uniformly approximates the corresponding PDE solutions over a parameter domain, under mild regularity assumptions.**
>
> Thank you for the clarification. While I think Theorem 4.3 could benefit from making this more explicit, the added clarification below aids the reader.

---

> > ### Author Response · Authors · 2026-01-24
> > **Response to Follow-up Comments**
> >
> > Thank you for taking the time to review the revised manuscript and for your follow-up comments. We appreciate your acknowledgment of the improvements to the manuscript. We have further revised the manuscript accordingly. Please see our response to your additional comment below.
> >
> >
> > **The results in Section 5.1 point to a benefit of the analytical approach over finite differences in training time. However, the impact of imposing Dirichlet boundary conditions remains unaddressed. An experiment demonstrating a practical benefit would strengthen the case, as fixed boundary conditions are prominently emphasized (e.g., in the abstract).**
> >
> > We thank the reviewer for this suggestion. We have further revised the manuscript by adding a new ablation study in Appendix E.3 to explicitly investigate the effect of direct ICBC assignment versus ICBC loss enforcement for PI-BSNet. Specifically, we compare the original PI-BSNet with direct ICBC assignment against a variant where ICBCs are imposed via an additional loss term for the convection diffusion equation experiment. The results show that directly assigning ICBC values through the B-spline formulation leads to lower prediction errors ($1.45 \pm 0.21 \times 10^{-2}$) compared to enforcing ICBCs via a loss penalty ($5.52 \pm 0.55 \times 10^{-2}$). While carefully tuning the additional ICBC loss together with the other loss terms can potentially yield improved performance, this process is nontrivial and often sensitive to hyperparameter choices [1, 2]. We hope these results help clarify the practical impact of direct ICBC assignment and provide additional supporting evidence.
> >
> > [1] Sifan Wang, Yujun Teng, and Paris Perdikaris. Understanding and mitigating gradient flow pathologies in physics-informed neural networks. SIAM Journal on Scientific Computing, 43(5):A3055–A3081, 2021a
> >
> > [2] Aditi Krishnapriyan, Amir Gholami, Shandian Zhe, Robert Kirby, and Michael W Mahoney. Characterizing possible failure modes in physics-informed neural networks. Advances in neural information processing systems, 34:26548–26560, 2021.

---

> > > ### Comment · Reviewer_qsJy · 2026-01-28
> > > **Response to Revision**
> > >
> > > Thank you for the further revision, I think now the proposed architecture is now properly motivated and ablated. From these revisions, I can correct my earlier assessment and affirm that the claims made in the submission are supported by accurate, convincing and clear evidence.

---

> > > > ### Author Response · Authors · 2026-01-29
> > > > **Response to Reviewer**
> > > >
> > > > We are pleased to hear that the revisions addressed your earlier concerns. Thank you again for your time and constructive feedback, which helped improve the quality of this work.

---

### Review · Reviewer_YT1J · 2025-12-02

**Summary Of Contributions:**

The paper examines the problem of approximating solutions to parametric PDEs, where parameters may influence the initial and boundary conditions, the differential operator, and even the geometry of the domain. To address this challenge, the authors propose Physics-Informed Deep B-Spline Networks (PI-BSNet), a framework that combines fixed B-spline basis functions with physics-informed learning to estimate the associated coefficients and approximate the PDE solution. The reported results demonstrate the potential of this model for solving parametric PDEs. The authors also provide theoretical guarantees showing that PI-DBNs act as universal approximators for certain classes of parametric PDEs and establish corresponding generalization-error bounds.

# Strengths:
* Representing solutions in a fixed B-spline basis and learning only the corresponding weights offers a more interpretable alternative to fully neural PDE solvers.
* A notable advantage of the proposed approach is that the B-spline construction enables the exact enforcement of initial and Dirichlet boundary conditions, thereby eliminating the need for explicit boundary-loss terms in the training objective.
* By embedding physics information into the B-spline basis, the method eliminates the need to compute higher-order neural network derivatives during training, then speeding up the training procedure and addressing a difficulty in PINNs and related physics-informed approaches.

# Weaknesses:

The structure of the paper is generally well organized, but several issues reduce clarity:
* Figure 1 is confusing, is it possible to make it more concise?,
* In the theoretical section, the introduction of the inner product and square integrability in the main body of the paper feels out of place, as these concepts are not actively used in the main paper.
* The use of norms is unclear, particularly in Assumption 4.4. Aren’t the functions $G_{\theta}(u,\alpha)$ vector-valued?
* The parameter spaces, $\mathcal{A}$ and $\mathcal{U}$ would be better introduced earlier in the paper to improve clarity.

The theoretical analysis depends on standard universal approximation results for neural networks. The contribution would be significantly strengthened by results that leverage the approximation properties and convergence rates of B-splines.
* Can you clarify the source of the claim that the solution spaces for any elliptic and parabolic PDE have the stated regularity? A reference would be helpful.
* Do you have a proof or a reference for the proof of Lemma A.6?
* Can you use equation 47 from the appendix and verify numerically that you can obtain the expected convergence rate for some PDE using your model and standard B-splines? I believe this may be more informative about the perfomance of the model.
* Is it possible to obtain figures showing that indeed the model realy follows the generalized error bounds for some PDE?

Although the method is meant to solve parametric PDEs, the experiments mostly show single-instance solutions, so it is hard to tell how well it works across different parameter values.
* It is unclear why only 10 and 30 test trials were conducted or how these trials were selected.
* A metric to measure the error should be chosen across the experiments, I found different metrics accross the experiments.
* Only one experiment compares the method to baselines, and some of those baselines are not really designed for parametric PDEs. To make a fair comparison, the baselines should be trained for their best performance, rather than just using similar architectures, since the models are fundamentally different.
 * It is mentioned that Neumann and Robin conditions still need loss-penalty terms. However, the paper doesn’t include any experiments showing how well the method handles these boundary conditions compared to other approaches.

**Additional Comments:**

N/A

**Audience:**

Yes

**Audience Explanation:**

This model tries to address a problem that lies at the core of Uncertainty Quantification (inverse problems, PDE-constrained optimization, etc.), even though the use of B-spline representations is not new. The model deserves merit for combining a well-established mathematical framework with machine-learning techniques in a way that may alleviate some of the limitations of classical methods.

**Claims And Evidence:**

No

**Claims Explanation:**

The experimental section does not fully demonstrate the parametric performance of the model. It is unclear why only 10 and 30 test trials were conducted or how these trials were selected. Additionally, there is no clear rationale for which metrics should be reported, as three different metrics are presented without explanation. Only one experiment compares the method to baselines, and some of those baselines are not really designed for parametric PDEs. To make a fair comparison, the baselines should be trained for their best performance, rather than just using similar architectures, since the models are fundamentally different.  Moreover, there are some details to be clarified about the theoretical part.

**Requested Changes:**

* Try to address the comments regarding the presentation of the paper.
* Report how the model behaves when using more B-splines or more data, or include results related to the convergence rates of the model (eq 47).
* Add an experiment demonstrating that the model follows the generalized error bounds for a representative PDE.
* Standardize the metrics used for the test trials.
* Compare the method against the strongest relevant baseline. PINNs are not designed for parametrized PDEs, so a more appropriate baseline should be included.
* Add an experiment evaluating the performance of the model under Neumann or Robin boundary conditions.

---

> ### Author Response · Authors · 2026-01-09
> **Author Response**
>
> Thank you for your thoughtful review and constructive feedback. Please find our detailed responses to your comments below.
>
> **Figure 1 is confusing. Is it possible to make it more concise?**
>
> We thank the reviewer for this suggestion. We have revised Figure 1 to make it more concise by removing the explicit B-spline basis formulas and optimized flow to improve readability.
>
> **In the theoretical section, the introduction of the inner product and square integrability in the main body of the paper feels out of place, as these concepts are not actively used in the main paper.**
>
> We thank the reviewer for this observation. To improve readability, we have moved the definitions of the inner product and square integrability to the Appendix, where they are directly used in the proofs.
>
> **The use of norms is unclear, particularly in Assumption 4.4. Aren’t the functions $G_{\theta}(u,\alpha)$ vector-valued?**
>
> We thank the reviewer for pointing out this ambiguity. In the revised manuscript, we have clarified the norm notation in Assumption 4.4 by explicitly stating the vector 2-norms. We have updated usage elsewhere as well to improve clarity.
>
> **The parameter spaces, $\mathcal{A}$ and $\mathcal{U}$, would be better introduced earlier in the paper to improve clarity.**
>
> We thank the reviewer for this suggestion. In the revised manuscript, we now introduce the parameter spaces $\mathcal{A}$ and $\mathcal{U}$ explicitly in the problem formulation (Section 3.1), where the PDE, domain, and associated parameters are first defined, to improve readability.
>
> **Can you clarify the source of the claim that the solution spaces for any elliptic and parabolic PDE have the stated regularity? A reference would be helpful.**
>
> We thank the reviewer for this question. We understand this comment to refer to Assumption 4.1, which assumes continuity of the PDE solution with respect to the parameters. We do not claim that this property holds for all elliptic and parabolic PDEs. Rather, Assumption 4.1 is a regularity assumption that is satisfied by many well-posed elliptic and parabolic PDEs with unique solutions. For example, linear Poisson, convection–diffusion, and heat equations with appropriate initial and boundary conditions satisfy such continuity properties [1].
>
> [1] François Treves. Fundamental solutions of linear partial differential equations with constant coefficients depending on parameters. American Journal of Mathematics, 84(4):561–577, 1962.
>
> **Although the method is meant to solve parametric PDEs, the experiments mostly show single-instance solutions, so it is hard to tell how well it works across different parameter values.**
>
> We thank the reviewer for this comment. We clarify that all experiments in the paper are conducted on parametric PDE families rather than single PDE instances, and the reported prediction errors are evaluated across the corresponding parameter spaces. To make this more explicit, in the revised manuscript we have added additional visualizations in Appendix D that show solution predictions for multiple representative parameter values.
>
> **It is unclear why only 10 and 30 test trials were conducted or how these trials were selected.**
>
> We thank the reviewer for this question. The test trials were selected by uniformly random sampling from the corresponding parameter spaces. The numbers of test trials (10 and 30) were chosen to test the parametric PDE prediction capability of the models and do not carry specific significance. All methods were evaluated on the same test sets to ensure a fair comparison. We have clarified this point in the revised manuscript.
>
> **Do you have a proof or a reference for the proof of Lemma A.6?**
>
> We thank the reviewer for this question. We have added a reference to Wang et al. [2, Theorem 5], which provides the corresponding generalization error bound for a non-parametric single PDE and directly supports Lemma A.6.
>
> [2] Zhuoyuan Wang, Albert Chern, and Yorie Nakahira. Generalizable physics-informed learning for stochastic safety-critical systems. IEEE Transactions on Automatic Control, 2025.

---

> ### Author Response · Authors · 2026-01-09
> **Author Response (Cont.)**
>
> **Can you use equation (47) from the appendix and verify numerically that you can obtain the expected convergence rate for some PDE using your model and standard B-splines?**
>
> We thank the reviewer for this insightful suggestion. We agree that equation (47) characterizes the approximation error of standard B-spline interpolation, but it does not directly apply to PI-BSNet, which additionally involves a learned coefficient network. For this reason, we do not claim that PI-BSNet strictly follows the convergence rate in equation (47). Instead, we include an ablation study in Appendix E.3 that empirically investigates the effect of the number of control points on PI-BSNet performance. As shown in Table 3 in the revised manuscript, the prediction MSE consistently decreases as the number of control points increases, which is in qualitative agreement with the universal approximation result in Theorem 4.3.
>
> **Add an experiment demonstrating that the model follows the generalized error bounds for a representative PDE.**
>
> We thank the reviewer for this suggestion. In the revised manuscript, we have added an experiment to empirically verify the derived generalization error bounds for a representative PDE. Specifically, in Appendix C.3, we conduct experiments on the advection equation and analyze the relationship between the maximum prediction error over a family of parametric PDE instances and the total training loss. The results show an approximately linear dependence between the maximum prediction error and the training loss, which is consistent with the theoretical generalization error bounds established in Theorem 4.6.
>
> **Standardize the metrics used for the test trials.**
>
> We thank the reviewer for this suggestion. In the revised manuscript, we have standardized the evaluation metric across all test trials by reporting the relative $L_2$ error for all methods and experiments.
>
> **Compare the method against the strongest relevant baseline. PINNs are not designed for parametrized PDEs, so a more appropriate baseline should be included.**
>
> We thank the reviewer for this comment. We agree that standard PINNs are not inherently designed for parametrized PDEs. In all of our experiments, we therefore implement PINN-based baselines using the commonly adopted approach of augmenting the network inputs with PDE and boundary-condition parameters, which allows PINNs to learn solution families in a reasonable and consistent manner. In addition, we include PI-DeepONet and PI-FNO as representative operator-learning baselines that are explicitly designed for parametrized PDEs. We clarify this experimental setup in the revised manuscript to better motivate the choice of baselines and ensure a fair comparison.
>
> **It is mentioned that Neumann and Robin conditions still need loss-penalty terms. However, the paper does not include any experiments showing how well the method handles these boundary conditions compared to other approaches.**
>
> We thank the reviewer for this comment. In the revised manuscript, we have added experiments on parametric PDEs with Neumann boundary conditions to explicitly evaluate the performance of the proposed PI-BSNet under such boundary conditions. Since B-splines do not support direct assignment of derivative boundary conditions, the Neumann boundary conditions are imposed via loss terms for PI-BSNet, consistent with the discussion in the main text. We compare PI-BSNet against PINN and PI-DeepONet under the same experimental settings and demonstrate that PI-BSNet achieves comparable accuracy while being significantly more efficient. These results are presented and discussed in Appendix C.4 of the revised manuscript, thereby demonstrating the efficacy of the proposed method for non-Dirichlet boundary conditions.

---

### Review · Reviewer_6L5p · 2025-12-30

**Summary Of Contributions:**

The paper introduces a new physics-informed machine learning model whose goal is to learn partial differential equations (PDEs) accurately with changing initial boundary conditions (ICBCs). The key parts of the proposed method is to use B-splines with control points on high-dimensional surface of interest. To deal with the boundary conditionals, a Dirichlet model is considered. The PS-BSNet model introduced is mainly characterised in Figure 1 and consists of two main elements, a fully connected NN (3 layers for example) that predicts control points and the B-spline approximation, whose gradients are tractable. The authors introduce two additional theoretical results, one focused on proving that the B-spline model for the context of physics-informed models, and another on the generalisation error guarantees. Finally, the empirical results show both the strenghts and weaknesses of the proposed approach against a good proportion of last-SOTA methods in the literature. Main proofs and theoretical results are somehow omitted of the main manuscript and kept in the Appendix.

**Additional Comments:**

N/A

**Audience:**

Yes

**Audience Explanation:**

N/A

**Claims And Evidence:**

Yes

**Claims Explanation:**

**Points of strength:**

I do think the paper is in good shape during large parts of the submitted manuscript. Sincerely, I actually liked the way PI-BSNet is designed and how it is flexible enough to combine the NN-modelled control points, the B-spline model, and the different constraints. Its adaptability to different dimensionalities and both on the range of the surface and the order of the PDE, looks like a significant merit. The connection with the rest of PINN models seems to be well drawn, at least for me, and not being entirely familiar with that particular subcommunity of works, the paper is understandable and the decisions taken meaningful.

The use and introduction of the B-spline model is actually important here, and I also like how the authors exploit its properties on the computational side. That's very cool. However, I miss a bit of discussion and more details on that side, as well as on the other. But I will add more details on this later on.

The two theoretical results are also significant, although I miss a bit the purpose of introducing them in that way. Let's say that they are not well integrated in the flow of details and descriptions. I understand more the importance here of the first one for the universal approximation idea, and the second one confuses me somehow.

The empirical results are not extremely long, but they are precise. I would have loved to have the same ones, perhaps a bit more extended, not on SOTA methods but on PDE examples or other types of benchmarks common in the PINN context.

**Points of weakness:**

All the strengths given and perceived from the PI-BSNet get some shadows from the way the three losses are combined in Eq. 11) with the temperature hyperparameters. From the very first moment this comes into play, details and discussion is missing. Being aware that such a combination is somehow the first decision that one can take (linear combination), I can also say this particular task can get into issues or lack of good fitting easily. Multimodal models have told us a lot in the past. How optimization and such combination of losses lead correctly the PI-BSNet to get a good solution is a mystery to me, and Figure 3 partially confirms that --- why the data loss dominates and outperforms while not the others?

The tractability of the B-splines is super interesting and important, as I said before. However, I miss the discussion on its scalability, the way it gets balanced with the NN for the control points, and how the computation of the backward pass scales with the number of control points, order of the PDE, and data points.

Last but not least, the theoretical results are somewhat out of context to me, and even if they provide important insights to this piece of work, they move the point of view away of PDEs, types of PDEs, families, and examples. From my perspective, I would've prioritized communication more on the side of providing insights on the type of PDEs, dimensionality, practical challenges, and issues of other PINNs that this model solves... instead of the universal approximation and the generalization. This last one does not really solve my concerns about the overfitting chances of using Eq. 11 under certain scenarios of data scarcity that are not super discussed. While going to the Appendix for checking the proofs on the first theorem, I had significant doubts and concerns about the potential existence of other similar B-spline results on the universality of the approximation. Is it possible that similar results are already there? I say this mainly bc the theorem and proofs are more based on the B-splines model rather than on the general PI-BSNet.

**Requested Changes:**

I would be glad to see a significant update on the following points, that have also been mentioned before in the previous sections:

- Discussion and details on the B-spline tractability and scalability of the forward pass.
- More examples and context of the particular PDEs that the work could focus on in the near future, and what are the main practical motivations that authors had for the design of PI-BSNet and the choice of solutions for the parabolic and exponential families.
- Discussion and details on Eq. 11, its role in the empirical results of Figure 3 and just explaining weaknesses or strengths of such an approach.
- Accommodating better the theoretical results in the manuscript, being better integrated and less being an appendix to the main thread of the work.
- Larger introduction, perhaps more practical and aligned with PINNs such as the work is really valued in that direction in the near future.
- Changes to have the reader better informed on the power of B-splines, what theoretical results are there already (for instance, on the universal approximation thing).

---

> ### Author Response · Authors · 2026-01-09
> **Author Response**
>
> Thank you for your thoughtful review and constructive feedback. Please find our detailed responses to your comments below.
>
> **Discussion and details on the B-spline tractability and scalability of the forward pass.**
>
> We thank the reviewer for this suggestion. In the revised manuscript, we have expanded the discussion on the tractability and scalability of the B-spline–based forward pass in Section 6. Specifically, we clarify that while the number of control points directly affects the training efficiency of PI-BSNet through the scaling of the coefficient network, the test-time forward pass remains computationally inexpensive and shows negligible dependence on the number of control points. We further support this discussion by adding forward-pass timing results in Table 3, demonstrating that the inference time is on the order of $10^{-4}$ seconds and is largely dominated by framework-level overhead rather than spline evaluation or neural network computation.
>
> **More examples and context of the particular PDEs that the work could focus on in the near future, and what are the main practical motivations that the authors had for the design of PI-BSNet and the choice of solutions for the parabolic and exponential families.**
>
> We thank the reviewer for this suggestion. In the revised manuscript, we clarify that the design of PI-BSNet is primarily motivated by practical problems such as risk quantification for unknown stochastic dynamical systems, where safety probabilities are governed by families of parabolic PDEs that must be evaluated repeatedly under changing dynamics, parameters, and ICBCs. This setting motivates a representation that can efficiently learn and evaluate families of PDE solutions rather than solving each instance from scratch.
>
> We also expand the discussion on future directions to note that PI-BSNet is not limited to the specific PDEs studied in this paper. In particular, future work may explore a broader class of PDEs whose solutions admit accurate B-spline representations, as well as extensions toward uncertainty quantification and improved interpretability of physics-informed learning enabled by the structured, coefficient-based spline representation.
>
> **Discussion and details on Eq. (11), its role in the empirical results of Figure 3, and an explanation of the weaknesses or strengths of such an approach.**
>
> We thank the reviewer for this comment. Equation (11) defines the total training objective as a weighted sum of the PDE, data, and ICBC losses, whereas Figure 3 reports these three loss components separately to illustrate their individual behaviors during training. In particular, Figure 3 shows that PI-BSNet achieves lower PDE and data losses compared to most baselines, and that it does not require an explicit ICBC loss term when ICBCs are directly assigned through the spline representation. The loss function in equation (11) follows the standard formulation used in physics-informed neural networks [1, 2], whose strengths, limitations, parameter choices, and known failure modes have been extensively studied in prior work [3, 4].  We have revised the manuscript accordingly to reflect those discussions and to improve clarity.
>
> [1] Maziar Raissi, Paris Perdikaris, and George E Karniadakis. Physics-informed neural networks: A deep learning framework for solving forward and inverse problems involving nonlinear partial differential equations. Journal of Computational Physics, 378:686–707, 2019.
>
> [2] George Em Karniadakis, Ioannis G Kevrekidis, Lu Lu, Paris Perdikaris, Sifan Wang, and Liu Yang. Physics-informed machine learning. Nature Reviews Physics, 3(6):422–440, 2021.
>
> [3] Sifan Wang, Yujun Teng, and Paris Perdikaris. Understanding and mitigating gradient flow pathologies in physics-informed neural networks. SIAM Journal on Scientific Computing, 43(5):A3055–A3081, 2021a.
>
> [4] Aditi Krishnapriyan, Amir Gholami, Shandian Zhe, Robert Kirby, and Michael W Mahoney. Characterizing
> possible failure modes in physics-informed neural networks. Advances in neural information processing
> systems, 34:26548–26560, 2021.
>
> **Accommodating better the theoretical results in the manuscript, being better integrated and less being an appendix to the main thread of the work.**
>
> We thank the reviewer for this suggestion. In the revised manuscript, we now highlight the key points and implications of the theoretical results at the beginning of the theoretical section. All assumptions and theorem statements are presented in the main text, while detailed proofs are moved to the appendix to improve readability and keep the theoretical contributions integrated with the main text.

---

> ### Author Response · Authors · 2026-01-09
> **Author Response (Cont.)**
>
> **Larger introduction, perhaps more practical and aligned with PINNs such that the work is really valued in that direction in the near future.**
>
> We thank the reviewer for this suggestion. In the revised manuscript, we have revised and expanded the Introduction to better align the paper with the PINN literature and to emphasize practical motivations. In particular, we explicitly highlight several key challenges in existing PINN-based approaches, including ICBC compliance, training efficiency and theoretical analysis for physics-informed learning with structured basis representations. We also add concrete application context, such as risk quantification for safety-critical stochastic dynamical systems, to motivate the need for efficiently learning families of PDE solutions with rapid online inference.
>
> **Changes to have the reader better informed on the power of B-splines, what theoretical results are there already.**
>
> We thank the reviewer for this suggestion. In the revised manuscript, we have clarified the theoretical foundations of B-splines and their role in the proposed framework. In particular, we explicitly discuss existing universal approximation properties of B-splines and explain how these classical results are leveraged in our proofs to establish universal approximation guarantees for PI-BSNet on families of parametric PDEs. We also expand the discussion in the conclusion to highlight additional well-known properties of B-splines, such as convex hull guarantees, smoothness, and noise-filtering behavior, and how these properties motivate future directions including uncertainty quantification, robustness analysis, and interpretable physics-informed learning.

---

### Decision · Action_Editor_TzkS · 2026-02-14

**Recommendation:** Accept as is

**Additional Comments:**

The rebuttal phase saw the authors being quite responsive and involved, and as far as I can tell they already answered the major points raised by the reviewers.
The paper is accepted as is.

**Audience:**

Yes

**Audience Explanation:**

Physics-informed neural nets are a staple in constraint-aware ML, and therefore of interest to the broader community of injecting/enforcing constraints in neural networks. This particular work, while a bit incremental on novelty, bring a possibly useful perspective (and tools to tinker with) to this community.

**Claims And Evidence:**

Yes

**Claims Explanation:**

The paper introduces a neural surrogate for parametric PDEs based on B-splines with control points on high-dimensional surfaces to deal with the parameters in changing initial conditions and boundary conditions. This nice idea and its direction within the scope of physics-informed ML was praised by reviewers.

The authors addressed several criticisms during rebuttal, ranging from the lack of clarification in differences between the proposed deep B-splines nets and other constrained-aware ML models (used also for PDEs) using splines; regarding notation and presentation as the length of the manuscript is considerable and symbols were not always properly defined; and the lack of discussion regarding the choice of hyperparameters such as combining the different losses and the relevance of the theoretical results proposed.

One last pending concern is the relevance in terms of novelty. I personally think that is more than sufficient and authors never overclaim it.